# Neural activity during a simple reaching task in macaques is counter to gating and rebound in basal ganglia–thalamic communication

Bettina C. Schwab[1,2], Daisuke Kase[3,4], Andrew Zimnik[5], Robert Rosenbaum[6], Marcello G. Codianni[7], Jonathan E. Rubin[4,7], Robert S. Turner[3,4]*

**1** Department of Neurophysiology and Pathophysiology, University Medical Center Hamburg-Eppendorf, Hamburg, Germany, **2** Technical Medical Center, University of Twente, Enschede, the Netherlands, **3** Department of Neurobiology, University of Pittsburgh, Pittsburgh, Pennsylvania, United States of America, **4** Center for the Neural Basis of Cognition, University of Pittsburgh, Pittsburgh, Pennsylvania, United States of America, **5** Department of Neuroscience, Columbia University Medical Center, New York, New York, United States of America, **6** Department of Applied and Computational Mathematics and Statistics, University of Notre Dame, South Bend, Indiana, United States of America, **7** Department of Mathematics, University of Pittsburgh, Pittsburgh, Pennsylvania, United States of America

* rturner@pitt.edu

**Data Availability Statement:** Data and code can be found at https://doi.org/10.5061/dryad.0cfxpnvxm.

## Abstract

Task-related activity in the ventral thalamus, a major target of basal ganglia output, is often assumed to be permitted or triggered by changes in basal ganglia activity through gating- or rebound-like mechanisms. To test those hypotheses, we sampled single-unit activity from connected basal ganglia output and thalamic nuclei (globus pallidus-internus [GPi] and ventrolateral anterior nucleus [VLa]) in monkeys performing a reaching task. Rate increases were the most common peri-movement change in both nuclei. Moreover, peri-movement changes generally began earlier in VLa than in GPi. Simultaneously recorded GPi-VLa pairs rarely showed short-time-scale spike-to-spike correlations or slow across-trials covariations, and both were equally positive and negative. Finally, spontaneous GPi bursts and pauses were both followed by small, slow reductions in VLa rate. These results appear incompatible with standard gating and rebound models. Still, gating or rebound may be possible in other physiological situations: simulations show how GPi-VLa communication can scale with GPi synchrony and GPi-to-VLa convergence, illuminating how synchrony of basal ganglia output during motor learning or in pathological conditions may render this pathway effective. Thus, in the healthy state, basal ganglia-thalamic communication during learned movement is more subtle than expected, with changes in firing rates possibly being dominated by a common external source.

## Introduction

The connection between the basal ganglia (BG) and one of its major downstream targets, the thalamus, has received increased attention recently [1; 2; 3; 4; 5] due to its role as a key pathway by which the BG can influence cortical function. It is well established that the BG-thalamic

**Funding:** This work was supported by the National Institute of Neurological Disorders and Stroke at the National Institutes of Health grant numbers R01NS091853, R01NS113817 (to RST) and R01NS070865 (to RST and JER), NSF awards DMS1516288 and DMS1724240 (to JER), the Center for Neuroscience Research in Non-human primates (CNRN), 1P30NS076405-01A1, NSF awards DMS-1654268 and Neuronex DBI-1707400 (to RR), and the Netherlands Organization for Scientific Research (NWO, NDNS+ grant 613.009.012). The funders had no role in study design, data collection and analysis, decision to publish, or preparation of the manuscript.

**Competing interests:** The authors have declared that no competing interests exist.

**Abbreviations:** BG, basal ganglia; BOTA, burst offset–triggered average; BTA, burst-triggered average; CA, caudate; CCF, cross-correlation function; CI, confidence interval; EMG, electromyography; GPe, globus pallidus-externa; GPi, globus pallidus-internus; ISI, interspike interval; LTS, low threshold spike; NHP, nonhuman primate; NR, no response; PTA, pause-triggered average; SCP, superior cerebellar peduncle; SDF, spike-density function; SEM, standard error of the mean; TCN, thalamocortical relay neuron; VA, ventral anterior nucleus; VLa, ventrolateral anterior nucleus; VLp, ventrolateral posterior nucleus; VMb, ventral medial-basal nucleus; VPM, ventral posterior-medial nucleus.

projection is composed of GABAergic neurons [6; 7; 8] that fire at high tonic rates of about 60 spikes/s in nonhuman primates (NHPs) at rest [9] and that this projection terminates densely on thalamocortical relay neurons (TCNs), as well as on GABAergic thalamic interneurons (in species in which these exist) [10; 11]. The fundamental mechanism by which the BG-thalamic pathway communicates task-related information, however, remains uncertain [2]. A long-standing and widely accepted theory states that pauses in BG-output activity "gate" the task-related activation of thalamus via disinhibition [12; 13; 14; 15; 16; 17]. More specifically, the high tonic firing rate of BG-output neurons normally prevents thalamic neurons from responding to excitatory inputs (e.g., from cortex), but a task-related pause in BG output, which typically extends over 100 ms or more, would act as a disinhibitory opening of the gate allowing a temporally coordinated task-related activation of thalamus. A competing theory hypothesizes that BG output may promote subsequent thalamic activation. Specifically, the low threshold spike (LTS) mechanism common to thalamocortical neurons [4; 18; 19; 20] may produce rebound bursts of thalamic activity following a cessation of transiently elevated inhibition from the BG (e.g., following task-related increases in BG-output activity) [4; 21; 22]. Both of these theories predict a tight temporal control of thalamic task-related responses by changes in BG output; more specifically, the latency of task-related changes in BG-output activity should lead by a short time interval the resulting responses in thalamus. The gating hypothesis, in addition, predicts an inverse relationship in the signs of task-related changes in BG and BG-recipient thalamus, with the incidence (i.e., relative frequency of occurrence) of task-related increases in BG output associated with a proportional incidence of decreases in activity in BG-recipient thalamus.

Others have suggested that the influence of BG output on thalamus is more subtle, primarily consisting of a modulation of thalamic activity, whereas cortex is the primary driver. This idea arose initially from the observation that inactivations within the output nucleus of the BG did not abolish task-related activity in the BG-recipient thalamus [23; 24]. That result was corroborated recently by Goldberg and Fee [25] using the songbird model. Goldberg and Fee [25] also substantiated two insights suggested previously from between-studies comparisons [26; 27]: first, that task-related increases in firing are far more prevalent than decreases both in BG-output neurons [28; 29; 30; 31; 32] and in BG-recipient thalamus [17; 33; 34]; and second, that the latencies of task-related activity in BG-output neurons [28; 31; 32] do not lead those of the BG-recipient thalamus but may in fact lag behind them [17; 33; 34]. Goldberg and Fee [25] did observe strong inhibitory effects of individual BG efferents onto thalamic neurons. In that study, the primary mode of BG-thalamic communication was an entrainment of thalamic spiking to the interspike intervals (ISIs) of BG efferents. A similar mechanism, if present in mammals, would provide a way for BG output to modulate the timing of thalamic spiking without requiring, as the gating and rebound models do, strict relationships in the timing and sign of task-related changes in firing. However, the high strength of the entrainment observed by Goldberg and Fee [25] is likely a product of the unique synaptic anatomy of this circuit in the songbird [35] that is not present in mammals [10; 36].

The BG-thalamic loop circuit devoted to skeletomotor function provides a well-defined anatomically segregated substrate for testing the theories outlined above. This circuit receives convergent input from motor and somatosensory cortical areas and projects back to the motor cortices by way of a monosynaptic projection from the globus pallidus-internus (GPi) to the ventrolateral anterior nucleus (VLa) [37; 38]. Experimentally induced perturbations of this circuit are known to impair the vigor, timing, and initiation of voluntary movement [27; 39; 40]. A substantial diversity of opinion remains, however, about the specific contributions that this circuit makes to motor control under normal physiologic conditions [37; 41; 42; 43; 44; 45]. A better understanding of the mechanisms that govern GPi-to-VLa communication is likely to

reduce that uncertainty. Other downstream targets of the BG, including areas of the brainstem and midbrain [46; 47], are also of great importance but are not investigated here.

Here, for the first time (to our knowledge) in NHPs, we studied single-unit activity sampled simultaneously from connected regions of GPi and VLa while animals performed a highly standardized reaching task. Contrary to predictions of the gating theory, movement-related increases in discharge were common in both GPi and VLa. Furthermore, VLa task-related changes tended to begin earlier than GPi changes, inconsistent with ideas of gating and rebound. The firing of simultaneously recorded cell pairs in GPi and VLa was weakly (if at all) correlated, in contrast to what would be expected from gating or rebound models; bursts and pauses in GPi spontaneous activity had similar weak effects on VLa firing. Hence, our results challenge the view that task-related activity in the BG-recipient thalamus arises under most or all conditions from gating or rebound-inducing signals transmitted from the BG. We use simulations to show how the strength of GPi-VLa communication can depend critically on the degree of anatomical convergence in the GPi-VLa circuit and the strength of spike synchrony within GPi. Thus, our results do not speak to the function of this pathway under other task conditions or behavioral states and, in fact, are in congruence with existing observations of effective thalamic inhibition by synchronized BG output. Our results suggest that, during performance of well-learned tasks in neurologically normal animals, temporal influences of GPi outputs on VLa activity are subtle and at most modulatory. Control over the timing and intensity of both pallidal and thalamic discharge may be dominated by other, possibly cortical, inputs.

## Results

### Basic approach, database, and activity at rest

We studied the single-unit activity of neurons sampled from connected regions of the GPi and VLa. The spiking activity of isolated single units was recorded from both areas simultaneously in macaque monkeys while the animals performed a two-choice reaction time reaching task for food reward [48; 49]. In short, animals were trained to hold their left hand at a home-position located immediately to the left of the animal at waist height. Two visual targets were located 14 cm apart at shoulder height on a vertically mounted response panel positioned 30 cm in front of the animal. After a randomized home-position hold period, one of the targets (chosen pseudorandomly trial-to-trial) was lit and the animal had to move the left hand swiftly to that target (distance: approximately 37 cm) and then hold there to obtain a reward. The right arm was restrained in a padded splint throughout a data collection session. Multiple microelectrodes or 16-contact linear probes were positioned acutely in the arm-related regions of both nuclei [50].

We used a combination of electrophysiologic techniques to ensure that recordings were obtained from anatomically connected regions of GPi and VLa. Consistent with previous observations [34; 51], the location of the VLa nucleus was distinguished from surrounding thalamic regions—and from the cerebellar-recipient ventrolateral posterior nucleus (VLp) in particular—by the presence of a short-latency inhibition of spiking in response to electrical stimulation of the GPi and the absence of short-latency excitation in response to stimulation of the superior cerebellar peduncle (SCP, Fig 1B). This electrophysiologic localization of the VLa/VLp border was performed during initial mapping studies prior to formal data collection. Across multiple parallel electrode tracks, thalamic neurons inhibited by GPi stimulation (yellow tick marks in Fig 1A, examples 1 and 2 in Fig 1B) were always encountered at locations that were dorsal and anterior to neurons excited by SCP stimulation (green tick marks in Fig 1A, example 4 in Fig 1B). In most cases a narrow border (<0.5 mm) separated the deepest GPi-inhibited neuron and the dorsal-most SCP-excited cell (Fig 1A), and the location of this

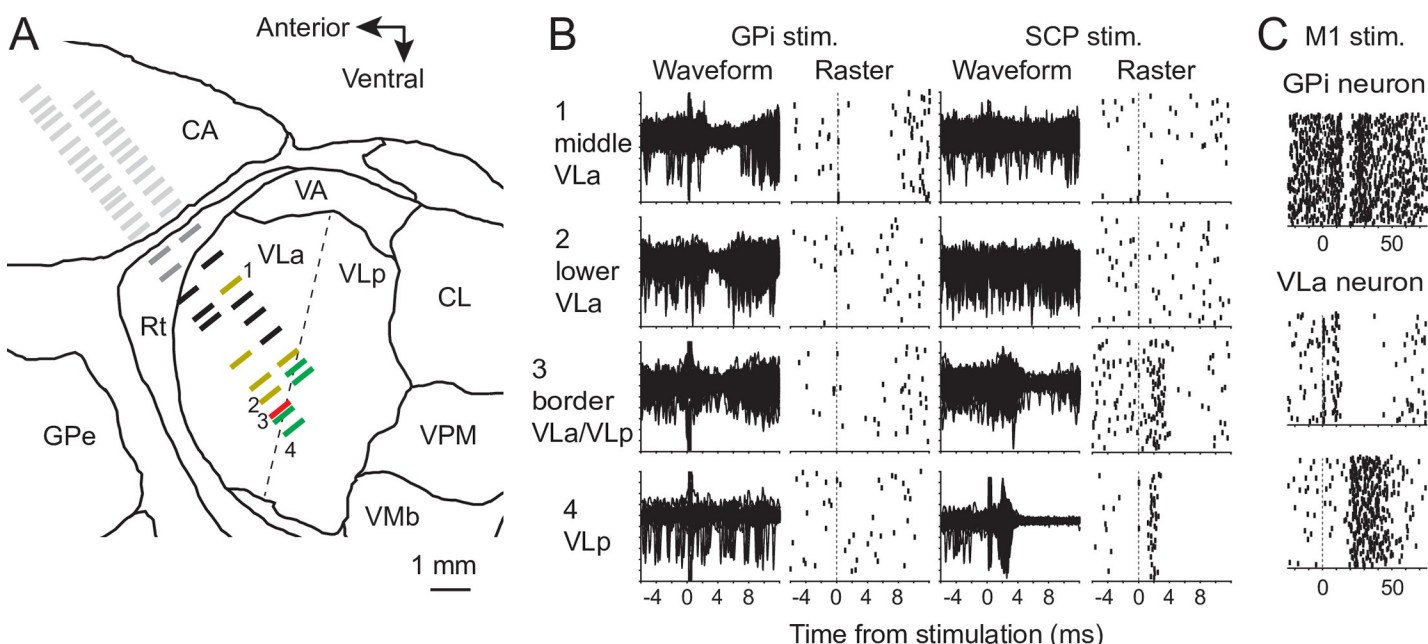

**Fig 1.** (A-B) Exemplar results from microelectrode mapping in the vicinity of the VLa thalamus. Single units encountered along parallel electrode trajectories were classified as being located in striatum (light gray tick marks), Rt (dark gray), or the VL thalamus. Neurons in VL thalamus were further classified as VLa neurons if they were inhibited by stimulation ("stim.") of the GPi and not excited by stimulation of SCP (yellow tick marks). Fig 1B1-2 shows overlaid peri-stimulus raw waveforms (left) and raster plots of sorted spikes (right) from two example locations in VLa (locations 1 and 2 in Fig 1A and 1B) at which GPi stimulation evoked a pause in neuronal activity and SCP stimulation had no effect. Neurons in VL were classified as VLp neurons if they were excited by stimulation of SCP and did not respond to stimulation of the GPi (green tick marks in Fig 1A; e.g., location 4 in Fig 1A and 1B). Neurons located at the border between VLa and VLp occasionally responded to stimulation of both GPi and SCP (red tick mark, location 3 in Fig 1A and 1B). Neurons that did not respond to stimulation were classified as VLa neurons (black tick marks Fig 1A) only if they were located antero-dorsal to the VLa/VLp boundary and postero-ventral to the Rt. (C) Regions of the GPi and VLa that belong to the arm-related BG-thalamic circuit were identified by testing for short-latency effects of electrical stimulation in arm-related areas of primary motor cortex. GPi neurons were included if sampled from regions at which stimulation of motor cortex (time zero) evoked a triphasic response at short latency (top, raster plot of sorted spikes). VLa neurons were included if sampled from regions at which stimulation of motor cortex evoked a pause or burst of activity at short latency (middle and bottom panels, respectively). BG, basal ganglia; CA, caudate; GPe, globus pallidus-externa; GPi, globus pallidus-internus; Rt, reticular nucleus of the thalamus; SCP, superior cerebellar peduncle; VA, ventral anterior nucleus; VLa, ventrolateral anterior nucleus; VLp, ventrolateral posterior nucleus; VMb, ventral medial-basal nucleus; VPM, ventral posterior-medial nucleus.

border aligned closely across adjacent electrode tracks. Coincident inhibition from GPi stimulation and excitation from SCP (e.g., Fig 1B3) was observed only at a small number of thalamic locations, and those were located exclusively in the narrow border between GPi-inhibited and SCP-excited regions (red tick mark in Fig 1A, example 3 in Fig 1B).

The subregions of GPi and VLa that compose the arm-related BG-thalamic circuit were identified by testing for short-latency effects of electrical stimulation in arm-related areas of primary motor cortex (Fig 1C). Previous studies have shown that stimulation of cortex can elicit complex triphasic responses in GPi neurons, via two- and three-synapse arcs through the BG [51; 52] (Fig 1C top), and that excitation of corticothalamic projections can elicit excitatory and inhibitory responses in VLa neurons [53] (Fig 1C bottom).

During subsequent data collection, thalamic single units were included in the database as VLa units if the electrode contact was located below the reticular nucleus of the thalamus (which is easily recognized neurophysiologically), above the VLa/VLp border (as defined above), and within 0.5 mm of a neuron affected by M1 stimulation. Single units sampled from the GPi were included if they were located in the dorsolateral portion of the nucleus and were encountered within 0.5 mm of a neuron affected by M1 stimulation. (See S1 Fig for anatomic locations of all included single units relative to observed electrophysiologic landmarks and inferred nuclear borders.)

**Table 1. Database and basic properties of neurons.**

| Measure | GPi | | | VLa | | |
|---|---|---|---|---|---|---|
| | NHP G | NHP I | Total | NHP G | NHP I | Total |
| *Number of units* | 105 | 104 | 209 | 66 | 119 | 185 |
| *Mean rate at rest (sp/s [SD])* | 70.8 (26.0) | 69.4 (29.9) | **70.1 (27.9)** | 14.8 (10.8) | 14.2 (11.4) | **13.5 (10.5)** |
| *Action potential width (ms min-to-max [SD])* | 0.24 (0.06) | 0.24 (0.11) | **0.24 (0.09)** | 0.46 (0.22) | 0.53 (0.19) | **0.50 (0.21)** |
| *Mvt-responsive neurons [number (%)]* | 102 (97) | 104 (100) | 206 (99) | 58 (87.9) | 105 (88.2) | 163 (88.1) |
| *Mvt-related responses* | 187 | 201 | 388 | 110 | 216 | 326 |
| *Increase only* | 109 | 80 | **189 (48.7)** | 81 | 132 | **213 (65.3)** |
| *Decrease only* | 33 | 37 | **70 (18.0)** | 20 | 57 | **77 (23.6)** |
| *Polyphasic increase first* | 29 | 37 | **66 (17.0)** | 5 | 13 | **18 (5.5)** |
| *Polyphasic decrease first (number [%])* | 16 | 47 | **63 (16.2)** | 4 | 14 | **18 (5.5)** |
| *Increases (proportion of all detected changes*)* | 66.4% (154/232) | 57.5% (164/285) | **61.5% (318/517)** | 75.6% (90/119) | 65.4% (159/243) | **68.8% (249/362)** |
| *Mean integrated change (sp [SD] × $10^3$)* | 4.38 (10.0) | 3.41 (11.4) | **3.89 (10.7)** | 3.24 (7.07) | 0.96 (3.61) | **1.75 (5.19)** |
| *Response latencies (ms relative to reach onset)* | | | | | | |
| *Increases (medians)* | −77.5 | −74 | −76 | −87.5 | −88.5 | −88 |
| *Decreases (medians)* | 31 | −52.5 | **−31.5** | −117 | −89 | **−95.5** |

**Bold**: significant difference between GPi and VLa neurons by Wilcoxon rank sum or chi-squared tests.

* For proportion of increases, biphasic responses are counted twice to account for the simultaneous presence of an increase and decrease in firing.

Abbreviations: GPi, globus pallidus-internus; max, maximum; min, minimum; Mvt, movement; NHP, nonhuman primate; sp, spikes; VLa, ventrolateral anterior nucleus

For analyses of task-related changes in activity, a total of 209 single units met the criteria to be included as GPi neurons and 218 as VLa neurons (Table 1). These units were studied over the course of 126 ± 82 trials of the behavioral task (mean ± SD; mean 63 trials for each of two movement directions; minimum number of trials per direction: 16; minimum duration of recording: 174.7 s). As expected, the resting firing rate of GPi neurons was significantly higher than that of VLa neurons (Table 1; $p = 1 \times 10^{-81}$, rank sum test) [34], and those rates and the differences between neural populations were highly consistent for the two animals (Table 1; $p = 1$, rank sum test). Also, as expected, the action potentials of GPi neurons were short in duration as compared with those of VLa neurons (Table 1; $p = 1 \times 10^{-33}$, rank sum test).

The mean firing rate of most single units showed small but significant ramps during the start-position hold period (i.e., before presentation of the task's go cue; $p < 0.05$, linear regression; 97% and 98% of GPi and VLa cells, respectively). This phenomenon is illustrated for exemplar GPi and VLa units in Fig 2A. The GPi unit's firing rate increased slowly (4.3 spikes/$s^2$, $p < 0.001$) during the approximately 1.2-s hold period before appearance of the go cue (red tick marks in rasters, Fig 2A) whereas that of the VLa unit decreased slowly (−6.7 spikes/$s^2$, $p < 0.001$; Fig 2A). For both GPi and VLa unit populations, the observed slopes were distributed symmetrically around zero (means: −0.05 and −0.14 spikes/$s^2$ respectively; $p = 0.29$, rank sum test). Positively and negatively sloped ramps in activity were equally common (49% and 51%, respectively), and those fractions did not differ significantly between GPi and VLa neurons or directions of movement ($p > 0.27$, chi-squared test). These linear trends in delay period activity were taken into account by the algorithm used to detect peri-movement changes in firing rate, as described in the following sections.

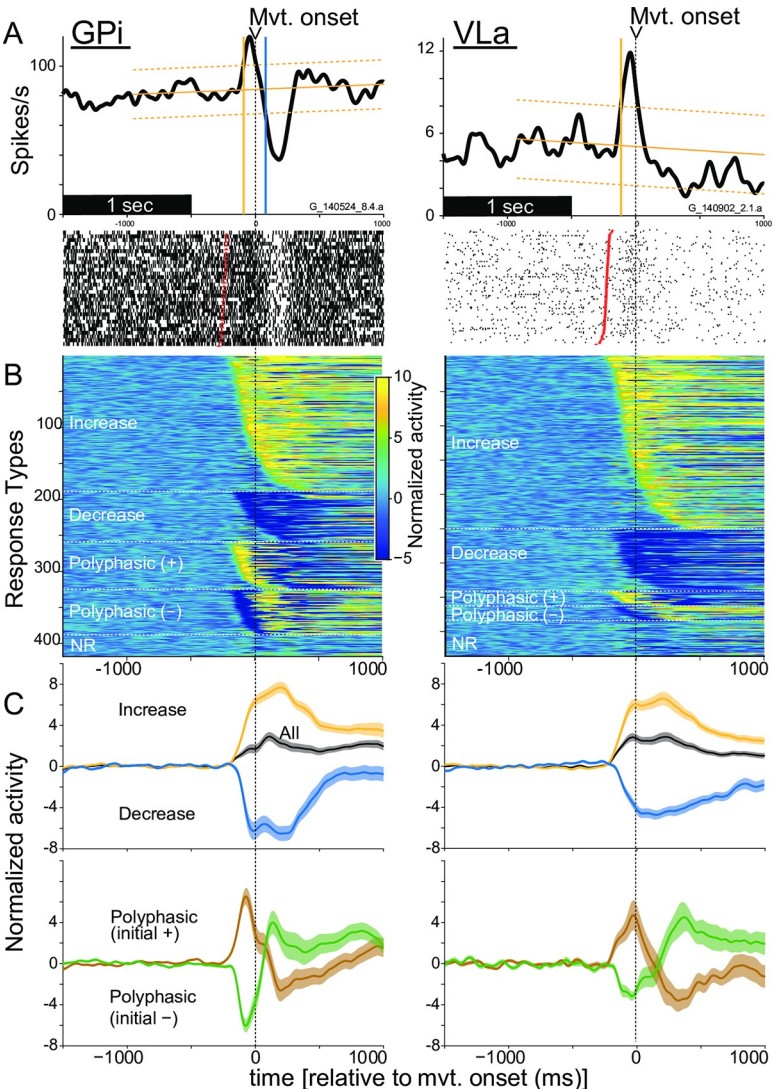

**Fig 2.** (A) Activity of one exemplar single unit sampled from GPi (left) and from VLa (right) aligned to the time of movement ("Mvt.") onset (vertical black dashed line). Peri-movement spike-density functions (top) and rasters (bottom) show highly consistent changes in discharge rate around the time of movement onset (time zero). Vertical yellow and blue lines: the times of onset of increases and decreases in discharge detected relative to that unit's baseline activity (sloping yellow trend line ± confidence interval). For raster plots, trials are sorted according to reaction time. Red tick marks: trial-by-trial times of go-cue presentation. (B) Spike-density functions of all single units studied sorted according to response form and response onset latency (earliest onsets at the top for each response form). Spike-density functions were z-scored relative to mean rate prior to go-cue presentation and displayed on a color scale. (C) Population-averaged spike-density functions for all GPi and VLa units (black) and for subpopulations with different response forms (as labeled). Shaded areas above/below the means reflect the SEM across units. Data and code to reproduce this figure can be found in https://doi.org/10.5061/dryad.0cfxpnvxm (Fig2_3_S8_S9.m). GPi, globus pallidus-internus; NR, no response; SEM, standard error of the mean; VLa, ventrolateral anterior nucleus.

## Task performance

Both animals performed the behavioral task in a highly stereotyped fashion with short reaction times and movement durations (S2 Fig). Reaction times did not differ significantly between the two animals (F[1,452] = 0, $p$ = 0.97, ANOVA) or between the two reach directions (F[1,452] = 0.2, $p$ = 0.68). There was a slight (7 ms), yet significant, difference between the two animals in the effect of target direction on reaction times (animal × direction interaction; F

[1,452] = 9.6, $p$ = 0.002]. Movement durations were longer for reaches to the more-distant, right target than to the left (F[1,453] = 416, $p = 4 \times 10^{-66}$). NHP G moved more slowly overall compared with NHP I (F[1,453] = 1,419, $p = 1 \times 10^{-141}$) and that slowing was more dramatic for the right target (animal × direction interaction; F[1,452] = 36.3, $p = 1 \times 10^{-9}$). Errors and outliers in task performance occurred at low rates in both animals (4.8% ± 1.2% and 3.4% ± 0.7% of trials in animals G and I, respectively; mean ± standard error of the mean [SEM]). Reach-related modulations in muscle activity (measured by electromyography [EMG], sampled during a subset of data collection sessions), were highly stereotyped across trials (S3 Fig). Importantly, reach-related changes in agonist muscle EMG began well before the overt onset of movement as detected mechanically and those onset latencies were similar for the two animals (−123 and −135 ms in NHP G and I, respectively; S3 Fig). The movement-related modulations in EMG differed quantitatively for the two directions of movement (widths of traces reflect SEMs in S3 Fig) but not in sign. The implications of these EMG results for the interpretation of single-unit activity are discussed in the following sections.

## Movement-related increases in firing are common in both GPi and VLa

We examined the peri-movement activity of neurons in GPi and VLa to test for evidence of an inverse relationship in their rate changes, as predicted by the gating hypothesis. In these analyses, responses for each direction of movement were considered separately. Large proportions of both neural populations modulated their firing rates around the onset of reaches to the left or right target (Table 1). We constructed mean spike-density and ISI functions for each single unit separately for movements to left and right targets and then tested for significant changes in firing rate relative to that unit's baseline activity (i.e., linear trend ± confidence interval [CI] of activity prior to go-cue presentation, yellow sloped lines in Fig 2A and S4 Fig). To avoid a bias inherent to spike-density functions (SDFs) toward detecting rate increases relative to rate decreases, we tested for peri-movement decreases in firing as increases in the mean ISI. (See S4 Fig for examples. See Materials and methods and S5 Fig for a simulation-based test for biases in the response detection algorithm used.) Fig 2A shows examples of the activity observed in individual GPi and VLa units. The GPi single unit showed a polyphasic change in firing that began with an increase in firing at −93 ms followed by a large decrease beginning at +77 ms relative to movement onset (yellow and blue vertical lines, respectively). The VLa single unit showed a monophasic increase in firing that first reached significance at −115 ms relative to movement onset. (The subsequent small but long-lasting depression in rate during movement did not reach significance.) Nearly all single units showed a significant peri-movement change in discharge for at least one direction of movement, with slightly fewer VLa units responding (99% of GPi neurons and 94% of VLa neurons; $p$ = 0.02, chi-squared test).

The form and timing of neuronal responses differed widely between neurons (see Fig 2B and S4 Fig) and, occasionally, between directions of movement (see paragraph beginning "The observations outlined above. . ." and S6A and S6B Fig). First, we considered neuronal responses independently for each movement direction and tested for differences between GPi and VLa populations in the incidence of different forms of peri-movement activity (see Table 1 and Fig 2B). Monophasic changes in firing (composed of a simple increase or decrease in firing) were the most common change detected in both GPi and VLa, amounting to 66.7% and 88.9% of all responses detected, respectively. Monophasic responses were more common in VLa than in GPi ($p = 2.8 \times 10^{-12}$, chi-squared test). Conversely, polyphasic responses—a series of one or more increase and decrease in firing—were more common in GPi than in VLa. Individual single-unit examples of each type of response are shown in S4 Fig, and all detected responses are shown sorted by response type and latency in Fig 2B.

Monophasic increases were the most common form of response both in GPi and in VLa (48.7% and 65.3% of responses detected, respectively; Table 1). When the individual phases of all detected responses were considered independently, increases in firing were also more common than decreases both in GPi and in VLa (61.5% and 68.8% of changes, respectively; $p = 1.2 \times 10^{-17}$, chi-squared test) with increases being nominally more common in VLa than in GPi ($p = 0.03$, chi-squared test). Simulations indicated that the observed high incidence of increases in both GPi and VLa was unlikely to be the byproduct of methodologic biases toward the detection of increases (see S5 Fig and Materials and methods).

Another way to compare the balance of task-related increases and decreases between GPi and VLa is to examine the mean firing rate across the two populations. Population averages combined across all responses types (all in Fig 2C) showed increases in firing rate during the peri-movement period for both cell types. This inclination toward increases was confirmed quantitatively by integrating changes in firing rate from baseline across the peri-movement epoch individually for each neuron. The mean of this integrated change was positive both for GPi and VLa (z = 7.37, $p = 1.7 \times 10^{-13}$, and z = 6.59, $p = 4.7 \times 10^{-11}$ for GPi and VLa, respectively, rank sum test; Table 1).

The observations outlined above were confirmed in analyses that considered potential influences of the direction of movement on unit discharge. Although the majority of GPi neurons (109 out of 206) and some VLa neurons (56 out of 179) showed significantly stronger responses to one of the movement directions, movement direction did not affect the magnitude or timing of peri-movement activity across whole GPi and VLa populations (S6A Fig; z < 1.76, $p > 0.07$, and z < 1.0, $p > 0.35$ for GPi and VLa, respectively, rank sum tests performed separately for each time point, uncorrected for multiple comparisons). The form of an individual neuron's peri-movement response (increase, decrease, or polyphasic) was typically consistent for reaches to the left and right targets (S6B Fig). This was true for both GPi neurons and VLa neurons (71% and 78% of neurons, respectively; shown by the color of bins along the matrix diagonals in S6B Fig; $p < 10^{-16}$, chi-squared test for homogeneity computed separately for GPi and VLa distributions). Again, increases were more common than decreases and polyphasic changes were more common in GPi than in VLa populations (compare diagonals between matrices for GPi and VLa in S6B Fig; $p = 2.9 \times 10^{-6}$, chi-squared test). We found no evidence for directional biases in the incidence of different forms of responses (e.g., decreases were not more common for one direction of movement over the other). A comparison of integrated changes in firing rate further reinforced these conclusions. For both GPi and VLa populations, the polarity and magnitude of a neuron's integrated change in firing for leftward movements was correlated with its change for rightward movements (S6 Fig. C; r > 0.6, $p < 1 \times 10^{-25}$ for both GPi and VLa populations). Notably, few points fell in the upper left and lower right quadrants of S6 Fig. C reflecting the fact that few neurons were modulated reciprocally by the two movement directions or exclusively for one direction. Simple conceptions of the gating model predict that such relations with movement direction would be common in GPi. The polarities of integrated changes (i.e., increase versus decrease from baseline rate) and the degree to which polarities agreed between movement directions were distributed very similarly for GPi and VLa populations (S6D Fig; $p = 0.68$, chi-squared test). However, despite this general similarity of individual units' responses for left and right reaches, peri-movement activity did differ significantly between left and right reaches for substantial fractions of GPi and VLa neurons (53% and 31% of units, respectively; S6C Fig; $p < 0.005$, rank sum test).

In summary, the general distribution of different response types and the predominance of firing-rate increases were similar in GPi and VLa (see S6E Fig), contrary to the prediction from the gating hypothesis that this relationship would be reciprocal. There was general similarity in the form and the sign of responses detected in GPi and VLa with the most notable

difference between populations being that polyphasic responses were more common in GPi than in VLa.

## Response onset latencies in GPi and VLa are incompatible with gating and rebound

The gating theory predicts that decreases in GPi activity should precede and permit increases in VLa firing, whereas a rebound mechanism would also feature changes in GPi activity that precede those in VLa. To address these predictions, we compared the times of onset for all individual changes in firing detected in GPi and VLa single units. As is evident in both Fig 2B and Fig 3A, response onset times were distributed widely across the peri-movement period, and that was equally true for neural responses in GPi and VLa. Quantitative comparison, however, showed that, on average, GPi changes in discharge began later than those in VLa (median onset times: −47 and −90 ms for GPi and VLa populations, respectively; z = 3.31, $p = 2.4 \times 10^{-5}$, rank sum test; Fig 3A), in contrast to the relationship posited in the gating and rebound frameworks. The more specific gating prediction states that decreases in GPi activity should precede VLa increases and, similarly, GPi increases precede VLa decreases. In contradiction, the distribution of GPi decreases lagged in time behind that of VLa increases (median onset times: −27 and −88 ms for GPi and VLa populations, respectively; z = 3.19, $p = 4.9 \times 10^{-4}$, rank sum test; Fig 3B). Similarly, VLa decreases did not follow but rather preceded GPi increases (median onset times: −62 and −108 ms for GPi and VLa populations, respectively; z = 2.58, $p = 0.01$, rank sum test; Fig 3C).

Although the overall difference in distributions for GPi and VLa response latencies is inconsistent with the gating hypothesis, the latency distributions do overlap substantially (Fig 3), leaving the possibility that a subpopulation of early-onset GPi responses begin earlier than later-onset VLa responses and thus could contribute to their generation (or engage in "gating"). To estimate how common such cases might be, we compared the timing and sign of every individual response detected in the whole database of GPi units against every response detected in the VLa database (regardless of whether or not GPi and VLa units were recorded from simultaneously) and computed the fraction of comparisons that conflicted with the gating hypothesis with respect to the mentioned parameters. A large preponderance of these comparisons (96%, $n = 178,846$ of 187,154 comparisons) directly conflicted with the gating hypothesis because of the relative timing or signs of the responses (i.e., GPi onset precedes VLa onset by 0–50 ms and response signs are opposite). Thus, when the apparent large overlap in latency distributions illustrated in Fig 3 is additionally constrained by predictions regarding response signs, the possibility for a gating-like latency relationship was found in only 4% of all possible GPi-VLa neuron pairs. The potential presence of an early-onset GPi subpopulation is addressed in more detail in the next sections.

The latency distributions described here did not differ significantly between the two directions of movement for neurons in either GPi or VLa (z < 1.19, $p > 0.08$, rank sum test). In addition, the latency estimation algorithms used here showed only slight nonsignificant biases when applied to simulated neuronal responses that matched the metrics of real GPi and VLa neuronal responses (z < 0.19, $p > 0.2$, rank sum test; S7 Fig; see Materials and methods). Moreover, the small biases observed in the simulation results were toward longer lags for the detection of VLa responses relative to the detection of GPi responses—the converse of what was observed in the empirical latency results. Qualitatively similar results to those from the general comparison of latencies (Fig 3) were also found when the latency analysis was restricted to the preferred direction of single units with peri-movement activity that was significantly directional (S8A–S8C Fig).

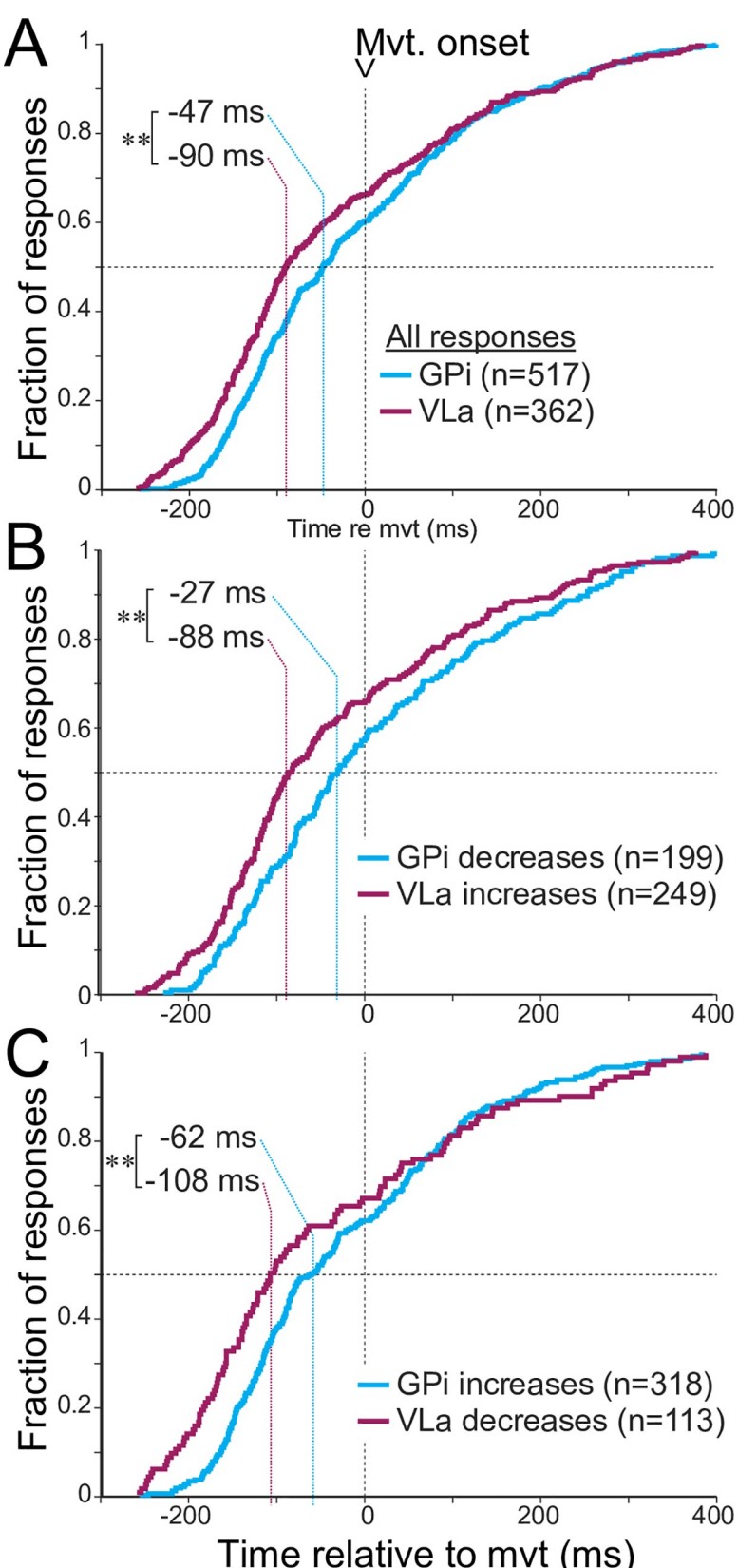

**Fig 3.** (A) Cumulative distributions of onset latencies of all significant peri-movement changes in firing detected in GPi neurons (blue) and VLa neurons (purple). Responses in VLa precede responses in GPi by a median of 43 ms ($^{**}p = 2.4 \times 10^{-5}$ rank sum test). (B) Response onset latencies of VLa increases (purple) lead GPi decreases (blue) by a median of 61 ms ($^{**}p = 4.9 \times 10^{-4}$ rank sum test). (C) Response onset latencies of VLa decreases (purple) lead GPi increases (blue) by a median of 46 ms ($^{**}p = 0.01$ rank sum test). Data and code to reproduce this figure can be found in https://doi.org/10.5061/dryad.0cfxpnvxm (Fig2_3_S8_S9.m). GPi, globus pallidus-internus; Mvt., movement; VLa, ventrolateral anterior nucleus.

As a further step to control for potential biases in latency estimation due, e.g., to differences between GPi and VLa populations in baseline firing rate and rate variability, we reestimated response onset latencies using an alternate approach based on the time at which an activity function crosses a fraction of the peak change (see Materials and methods). That approach resulted in slightly earlier-onset latencies overall (median latency shift: −14 ms), but the differences in latencies between GPi and VLa populations were fully consistent with those described above for the standard analysis (see S8D–S8F Fig). Thus, to summarize, the timing of changes in peri-movement discharge is not consistent with the idea that GPi activity triggers or gates task-related activity in VLa.

Finally, we looked for evidence that a subpopulation of early-onset GPi responses might lead and thereby gate subsequent VLa responses. For example, one might predict from the gating hypothesis that the magnitude of a GPi response is associated closely with the size of the response that it evokes milliseconds thereafter in VLa. Thus, GPi and VLa populations should show similar relationships between response magnitude and response latency if any such relationship existed. We therefore tested for relationships between response magnitude (measured here as mean change from baseline firing rate) and latency for all increases and decreases detected in GPi and VLa (S9 Fig). Contrary to predictions, however, the observed relationships differed markedly between GPi and VLa. In VLa, early latency responses were larger in magnitude than late onset responses as evidenced by the presence of significant negative correlations between latency and magnitude for both increase- and decrease-type VLa responses (Spearman $p < 0.05$) and significant 1-way ANOVAs across latency quartiles (F[3,248] = 4.3, $p = 0.005$ and F[3,112] = 4.3, $p = 1 \times 10^{-7}$, respectively). In GPi, however, no such relationship was found for increase-type responses (Spearman $p = 0.95$; F[3,308] = 1.5, $p = 0.2$), and nominally the opposite was found for decrease-type responses (Spearman $p = 0.65$; F[3,198] = 3.0, $p = 0.03$). In sum, the relation of response magnitudes and latencies does not support rebound- or gating-like mechanisms.

We next investigated the possibility that GPi-to-VLa communication might be evident in the activity of cell pairs sampled simultaneously from the two structures.

## Correlated activity in GPi-VLa cell pairs is rare and unbiased

Both gating and rebound triggering of thalamic activity by the BG should produce strong correlations in the precise timing of spikes in GPi and VLa. Contrary to that expectation, we found little evidence for strong short-time-scale spike-to-spike interactions between GPi and VLa unit pairs (Fig 4A–4F). When we computed cross-correlation functions (CCFs) for pairs of spike trains sampled simultaneously from GPi and VLa, only 3.5% of the CCFs showed any statistically significant modulation (15 of 427 cell pairs; $p < 0.05$ relative to a 20-ms-jittered control; Table 2). Similarly, small fractions of CCFs reached significance when the analysis was restricted to a rest period during the start-position hold period (2.2% of pairs; 4 out of 183), or to the peri-movement period (5.2% of pairs; 9 of 172; Table 2). For the movement period, the fraction of significant correlations did not increase when correlations were computed separately for the two movement directions (4.1% of pairs; 14 out of 342), and there was no

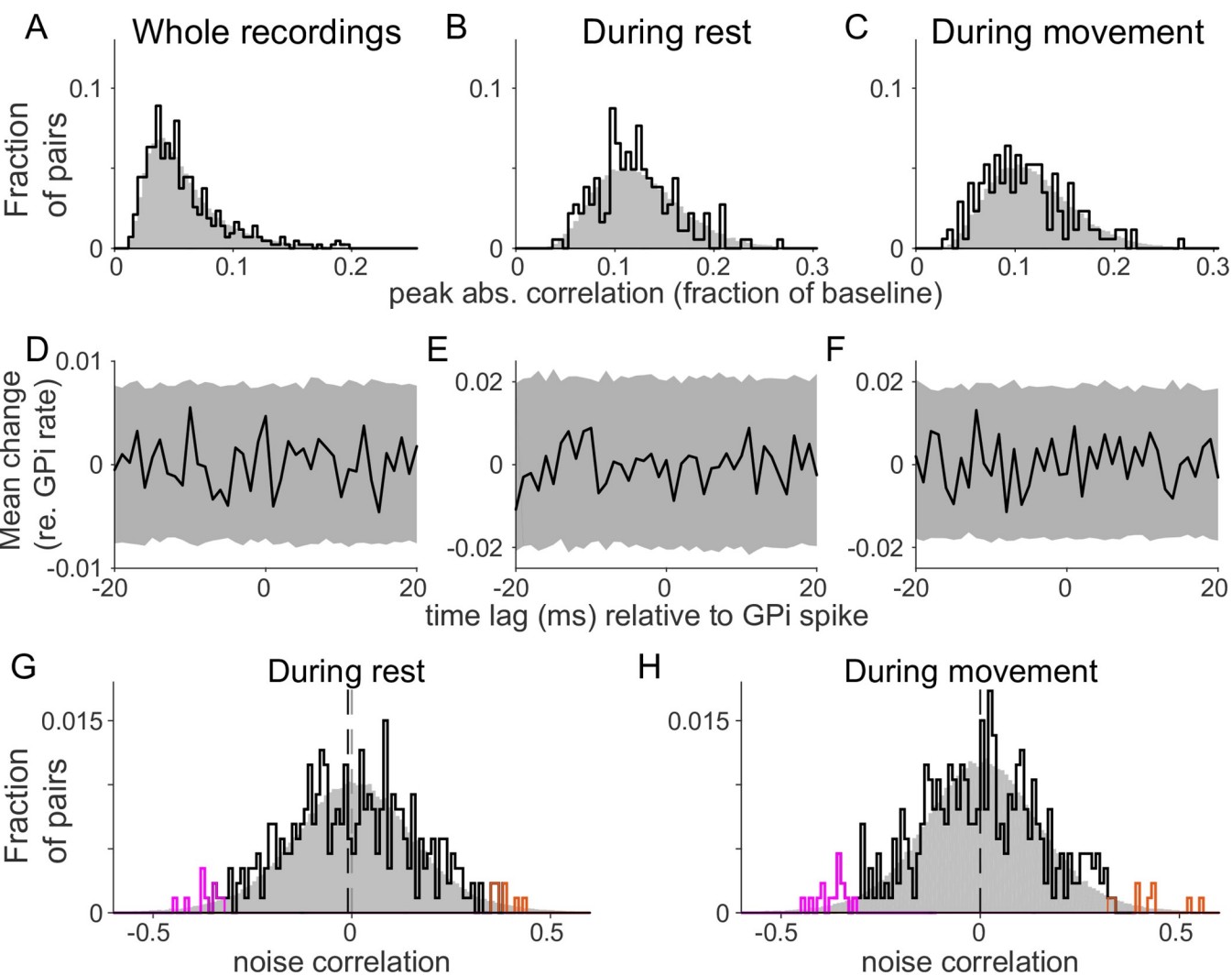

**Fig 4.** Absence of fast correlations between GPi and VLa spikes. (A-C) Histograms of peak absolute ("abs.") CCF values between GPi and VLa units (black) for whole recordings (A), during rest (B) and during movement (C). Jitter control distributions (20-ms jitter intervals) are shown in gray. (D-F) Population average CCFs (black) and 95% confidence intervals (gray) based on the jitter control. (G-H) Low noise correlations between GPi and VLa during both rest (G) and movement (H). Noise correlations exceeding the 95% confidence limits are shown in orange (positive) and magenta (negative). Dashed vertical lines indicate the medians of the distributions. For both rest and movement, test distributions did not differ significantly from the control distribution (gray). No significant bias toward positive or negative correlations was found. Data and code to reproduce this figure can be found in https://doi.org/10.5061/dryad. 0cfxpnvxm (Fig4_S10to13.m). CCF, cross-correlation function; GPi, globus pallidus-internus; VLa, ventrolateral anterior nucleus.

evidence that the occurrence of significant correlations was greater for one direction of movement over the other (6 out of 171 versus 8 out of 171; $p = 0.59$, chi-squared test). (Note that the number of pairs differs between time periods considered due to the strict selection criteria used to ensure adequate statistical power, as described in more detail in Materials and methods. The lower incidence of significant correlations in the direction-specific analysis may be related to the reduced number of trials.) Furthermore, the overall distribution of the peak absolute correlation values was not greater than the distribution of control peak absolute correlations taken from CCFs generated after jittering spike times within 20-ms time windows (Fig 4A–4C, uncorrected $p$-values: 0.975 for whole recordings, 0.675 for rest, and 0.998 for movement periods; one-sided permutation test). For none of the time periods considered did the rate of occurrence of significant correlations exceed by a significant degree the 5% rate of

**Table 2. Cell-pair interactions between GPi and VLa.**

| | Cross-correlations | Noise correlations | Burst/pause influences |
|---|---|---|---|
| *Number of GPi-VLa pairs* | Whole recordings: 427<br>During rest: 183<br>During movement: 172 | During rest: 332<br>During movement: 372 | Burst onsets: 163<br>Burst offsets: 167<br>Pause onsets: 153 |
| *Fraction of significant pairs (total number)* | Whole recordings: 3.5% (15) | During rest: 5.4% (18)<br>44% positive (8)<br>56% negative (10) | Burst onsets: 11.6% (19)<br>74% decreases (14)<br>26% increases (5) |
| | During rest: 2.2% (4)<br>During movement: 5.2% (9) | During movement: 7.2% (27)<br>41% positive (11)<br>59% negative (16) | Burst offsets: 7.8% (13)<br>77% decreases (10)<br>23% increases (3)<br>Pause onsets: 8.5% (13)<br>69% decreases (9)<br>31% increases (4) |

Significant cross-correlations: pairs with absolute CCF peaks exceeding 95% of the control distribution.

Significant noise correlations: pairs with maximum/minimum correlations larger/smaller than 97.5% of the control distribution.

Significant burst/pause influences: pairs with average VLa firing rates 0–100 ms after GPi burst/pause onset/offset above/below 97.5% of their baseline firing.

Abbreviations: CCF, cross-correlation function; GPi, globus pallidus-internus; VLa, ventrolateral anterior nucleus

false positives detected in the control distributions. More broadly, population averages of the CCFs did not extend beyond the CIs from the control data (Fig 4D–4F, $p > 0.05$ relative to shuffled control).

Among the small number of individual CCFs that did exceed the threshold for significance, none showed features consistent with being the product of strong monosynaptic GPi-to-VLa inhibition. Contrary to the expectation that GPi-to-VLa CCF effects would be predominantly inhibitory (i.e., negative), roughly equal numbers of the CCF peaks detected as significant were positive and negative (8 versus 7, respectively; $p = 0.79$, chi-squared test; S10D–S10F Fig). Only two CCFs showed a long-lasting (>5 ms) reduction in VLa firing following the GPi spike (S10D Fig, first and third examples). Those two effects showed features consistent with a slow rhythmic GPi-VLa synchronization at 25 Hz (approximately 40-ms cycle length). Slow rhythmic synchronizations at frequencies <25 Hz were also present in some individual raw CCFs, but those effects were removed implicitly by subtracting the average jitter control CCF (see Materials and methods). The magnitude of individual CCF peaks correlated closely with the width of the CCF's 95% CI (whole recordings: r = 0.82, $p < 10^{-10}$; during rest: r = 0.62, $p < 10^{-10}$; during movement: r = 0.61, $p < 10^{-10}$), which reflects how noisy the estimate of the CCF was (S10G–S10I Fig). In other words, the narrower the 95% CI, the smaller the estimation noise and, incidentally, the smaller the detected CCF peak. This was true even for CCF peaks that were detected as significant (red X's in S10G–S10I Fig), thus reinforcing the view that the CCF peaks detected here as significant may in fact have been false-positive noise events. Together, these observations suggest that short-time-scale spike-to-spike interactions between GPi and VLa unit pairs, if present, were too small and/or too uncommon to be detected with the methods used here.

Given these results, it was important to consider the sensitivity of our methods. To that end, we estimated how small of a known, experimenter-imposed, spike-to-spike correlation could be detected reliably in simulated spike-train pairs that matched the firing rates and recording durations of each empirical GPi-VLa pair. We also computed the upper bounds to each measured correlation (see Materials and methods for details). (These two measures were

distributed very similarly within each analysis time period [whole recordings, rest, or movement; S11D–S11F Fig].) Not surprisingly, the minimum reliably detectable correlation (our measure of sensitivity) varied across the population of simulated pairs because of between-pairs differences in firing rates and duration of data collection. This led to a trade-off between the number of pairs considered and the within-pair sensitivity for detection of small correlations. Considering whole recordings, it was possible to detect correlations smaller than 20.1% of baseline rate in 90% of pairs ($n = 388$) and correlations smaller than 10.5% in 50% of pairs ($n = 216$) (S11A Fig). In the most sensitive 10% of pairs, it was possible to detect simulated correlations smaller than 6.1% of baseline. Even in this last subset, in which real correlations should have been most easily detected, the number and magnitude of empirically detected CCF peaks did not exceed what would be expected if the CCFs were composed of noise alone. Analyses applied to rest and movement epochs lead to similar conclusions (S11B and S11C Fig). Again, the magnitude of individual CCF peaks correlated closely with the minimum reliably detectable correlation (whole recordings: r = 0.98, $p < 10^{-10}$; during rest: r = 0.97, $p < 10^{-9}$; during movement: r = 0.98, $p < 10^{-10}$), reflecting the noise in CCF estimation (S11D–S11F Fig), for all as well as for significant CCFs (red X's).

In sum, short-latency cell-to-cell interactions were too small and/or too uncommon to be detected reliably using the approach applied here. These observations bring into question not only the standard gating and rebound hypotheses but also more fundamental assumptions about the mechanisms at play in GPi-VLa communication. We addressed this issue in more depth using a computational approach (see Simulations. . . section).

Next, we tested for slow trial-to-trial correlations in firing rate ("noise correlations") between simultaneously recorded GPi-VLa cell pairs. This analysis was performed separately for rest and peri-movement periods. If gating was a common mechanism in GPi-VLa communication, then the majority of significant noise correlations would be expected to be negative whereas some versions of the rebound mechanism predict more frequent positive correlations due to the capacity of brief increases in GPi activity to effectively recruit prolonged rebound-supporting currents in VLa [4; 18; 19; 20]. We found that the overall distribution of noise correlations was not significantly different from the control distribution (Fig 4G and 4H; $p = 0.07$ during rest, $p = 0.388$ during movement; permutation test). Nevertheless, noise correlations did reach significance during rest for 5.4% of cell pairs (18 of 332 pairs; $p < 0.05$ relative to shuffled controls; Table 2) and during movement for 7.2% of cell pairs (27 of 372 pairs). Roughly equal fractions of those significant correlations were positive (orange lines, Fig 4G and 4H) and negative (magenta lines; 44% versus 56% during rest and 41% versus 59% during movement, respectively; Table 2). Also, between the recorded GPi neurons, cross-correlations and noise correlations were low (see S12 Fig). Examples of the largest three GPi-VLa noise correlations during movement are shown in S13 Fig. There was no evidence that the significant effects showed a bias in prevalence toward positive or negative correlations ($p = 0.64$ during rest, $p = 0.34$ during movement; chi-squared test).

Thus, neither of the tests for correlated activity in GPi-VLa cell pairs yielded evidence consistent with the gating or rebound hypotheses.

## VLa activity decreases with bursts and pauses in GPi activity

Bursts and pauses are common features of GPi activity that could facilitate the transmission of information to VLa. The postsynaptic effects in VLa of GPi bursts and pauses could also be larger in magnitude and easier to detect than effects produced by single GPi spikes. We therefore estimated the influences of bursts and pauses in GPi unit activity on the firing rate of VLa neurons during the rest period (Fig 5). The gating hypothesis predicts that GPi bursts should

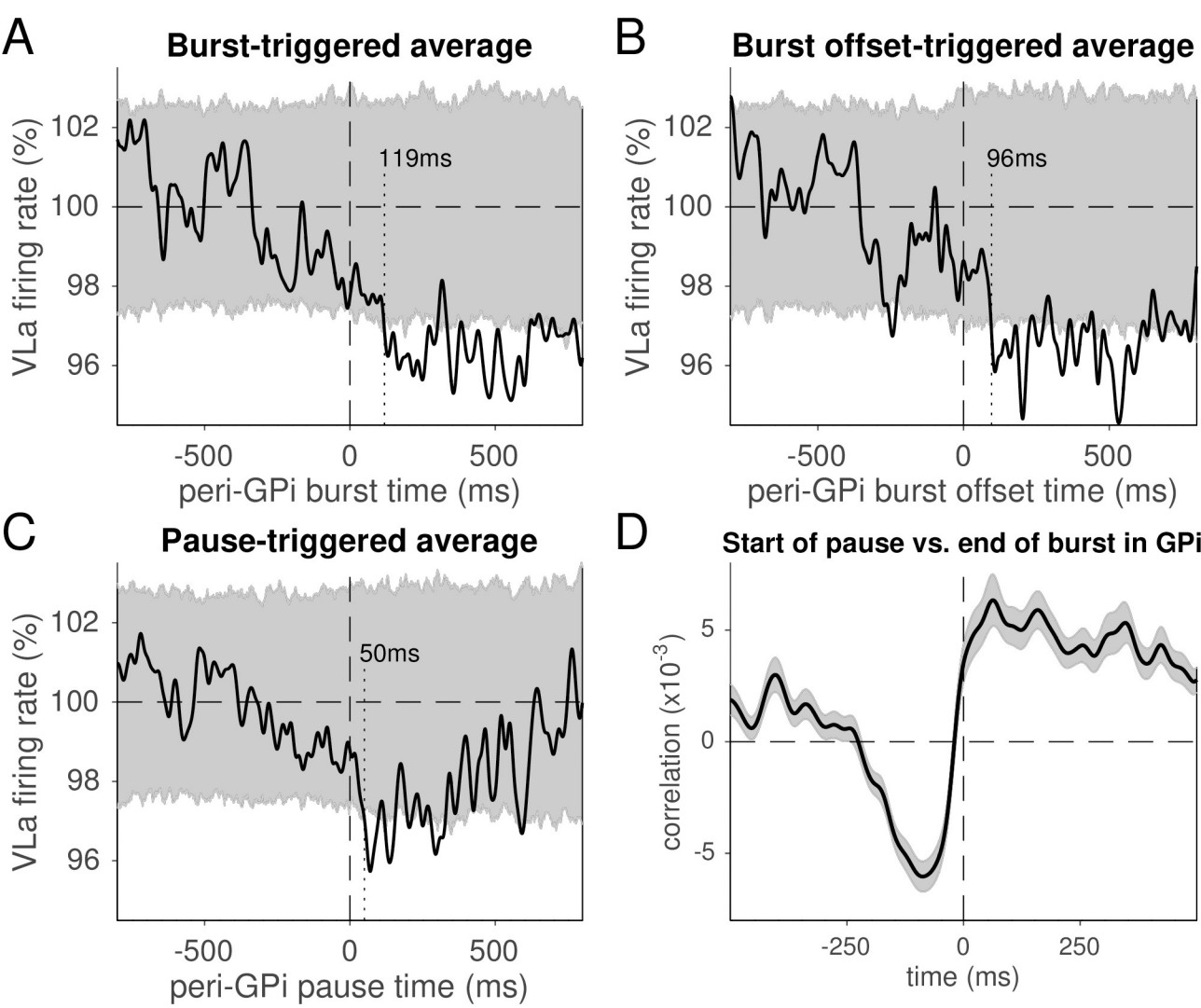

**Fig 5. VLa activity relative to GPi bursts and pauses.** (A-C) Population average VLa firing rates relative to GPi burst onsets (A), burst offsets (B), and pause onsets (C). The 95% confidence interval is shown in gray. In all three cases, VLa firing rates drop and stay low for several hundred milliseconds. (D) Population average cross-correlation of GPi pause onset times relative to the times of GPi burst offset (mean ± SEM). Pauses in GPi activity were more likely to occur immediately following the offset of a GPi burst. Data and code to reproduce this figure can be found in https://doi.org/10.5061/dryad.0cfxpnvxm (Fig5_S14.m). GPi, globus pallidus-internus; SEM, standard error of the mean; VLa, ventrolateral anterior nucleus.

be associated closely in time with reductions in VLa firing rate and GPi pauses with VLa increases. The rebound hypothesis predicts that GPi burst offsets should be followed by increases in VLa firing. We detected the occurrences of bursts and pauses in each GPi unit's rest period activity separately using standard methods [54] and then averaged the firing rates of simultaneously recorded VLa neurons around the times of GPi burst onset, burst offset, and pause onset. The resulting burst-triggered averages (BTAs), burst offset–triggered averages (BOTAs), and pause-triggered averages (PTAs) of VLa activity were averaged across the populations of qualifying GPi-VLa pairs (Fig 5A–5C; Table 2; see Materials and methods for selection criteria and S14 Fig. for an example GPi BOTA VLa firing rate from one pair). Small but significant transient decreases in VLa population spike rate were evident following both the onsets and offsets of bursts. That decrease began at a longer lag following the onset of bursts (119 ms) than after their offsets (95 ms) with the difference (24 ms) equal to the observed

mean duration of GPi bursts (24 ms). The presence of a decrease in VLa firing rate following the offset of GPi bursts is not consistent with the predictions of either gating or rebound hypotheses.

Surprisingly, a deceleration in VLa activity was also associated with pauses in GPi activity (Fig 5C). The VLa decrease reached significance 50 ms after GPi pause onset. Given that GPi pauses had a mean duration of 195 ms, the decrease in VLa firing rate occurred during GPi pauses, not following them. Population-averaged cross-correlations of the times of pauses in GPi unit activity relative to burst offsets for the same unit (Fig 5D) showed a broad (>200 ms) period of negative correlation (i.e., reduced likelihood of a pause) at negative time lags, as would be expected because pauses in firing are unlikely to occur during bursts. The probability of a pause swung sharply to positive values at the time of burst offset and remained positive for >500 ms thereafter. Thus, bursts and pauses in GPi activity tended to occur together in that order, with decreases in VLa firing rates occurring following GPi bursts, during GPi pauses.

An analysis of individual GPi-VLa cell pairs provided results consistent with the population-level burst/pause analysis. Only small fractions of GPi-VLa pairs showed any significant change in VLa activity following GPi burst onsets, burst offsets, or pause onsets (Burst/pause influences; Table 2). Among those significant effects, decreases in VLa firing rate were far more common than increases, composing more than two-thirds of the significant effects for all events (Burst/pause influences; Table 2). Because of the small number of cases, however, those differences in prevalence were only nominally significant and only so for burst onset and offset ($p$ = 0.04, 0.05, and 0.16 for burst onset, burst offset, and pause onset, respectively; chi-squared test).

Together, these results suggest that the activity of some GPi-VLa neuron pairs is coordinated such that burst-pause complexes in the GPi neuron's activity are associated with long-lasting reductions in the firing rate of the VLa neuron. The characteristics of this phenomenon, however, are not consistent with predictions of either gating or rebound models.

## Simulations suggest GPi-VLa communication scales with GPi synchrony and convergence

Given that GPi neurons project to VLa, it would be surprising if this pathway did not serve a functional role. Because we observed little impact of GPi activity on VLa in our empirical data, we turned to computational simulations to explore possible relationships between spiking in GPi and VLa (Fig 6). Specifically, we computationally generated sets of $N$ GPi spike trains, with pairwise correlations c, for various choices of $N$ and c. These $N$ spike trains provided converging inhibitory inputs to a single model thalamocortical relay cell in which synaptic conductances were normalized by $1/N$ to maintain a constant average synaptic conductance across different choices of $N$ (Fig 6A). (Homeostatic normalization of synaptic inputs similar to this is common in thalamocortical neurons [55].) Cross-correlations between spike times from randomly selected GPi input trains and the VLa neuron were computed for lags in the interval [−100 ms, 100 ms], and averages from 20 GPi neurons were collected over four separate runs for each ($N$,c) pair. Results showed a clear growth of the post-spike inhibition of VLa with increasing c (GPi pairwise correlation) when the number of converging GPi neurons ($N$) is held constant (Fig 6B). Importantly, when the within-GPi correlation is held constant, the post-spike inhibition of VLa decreased in magnitude as the number of converging GPi neurons ($N$) increased. Average correlations in the interval [0 ms,10 ms] were approximately proportional to $[(N - 1)/N]c + 1/N$ as predicted on theoretical grounds (Fig 6C; see also Materials and methods). For large $N$ and small c, the relative influence of individual GPi spike trains on VLa spiking became small owing to a large number of GPi spike trains contributing

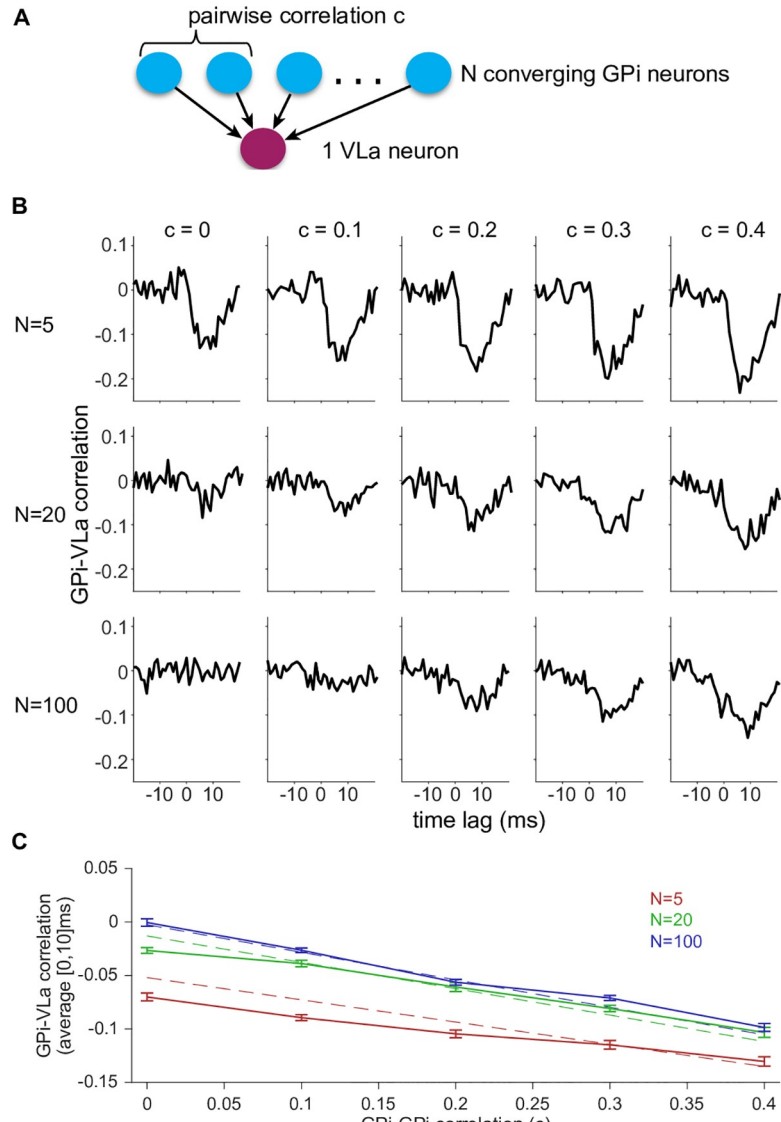

**Fig 6. GPi synchrony and anatomical convergence determine correlations between GPi and VLa.** (A) Computational model architecture. One VLa neuron is inhibited by $N$ GPi neurons with pairwise correlation c. (B) Population average cross-correlations across ($N$,c) combinations. (C) Average correlations in the interval [0 ms, 10 ms] are similar to a linear approximation (dashed lines; see Materials and methods for details). Bars indicate the standard error of the mean of the 20 trials. Data and code to reproduce this figure can be found in https://doi.org/10.5061/dryad.0cfxpnvxm (Fig6.m). GPi, globus pallidus-internus; VLa, ventrolateral anterior nucleus.

uncorrelated variability to the VLa membrane potential. Given a certain fixed amount of inhibition to each VLa neuron, these results highlight the importance of the degree of anatomical convergence from BG-output neurons onto recipient thalamic neurons, which is reported to differ markedly between homologous circuits in the songbird (in which $N$ = 1; [25; 35]) and mammals (in which $N$ may exceed 20). They also suggest that in contexts in which pairwise GPi cross-correlations are elevated (e.g., in parkinsonism [56; 57]), GPi outputs may exert a strong influence on subsequent VLa activity. Significant pairwise correlations in GPi spiking are rare, however, under baseline behavioral conditions in neurologically normal animals [56; 58; 59]. We confirmed that observation in the current data set by finding only 13 significant

spike-to-spike correlations among 148 pairs of GPi neurons examined (S12A Fig). Out of these 13 significant correlations, three were classified as strong outliers (>10 × median absolute deviation away from the population median). Although the mean of all CCFs just exceeded the threshold for significance (S12D Fig, black line), that significance was lost after removal of those three outliers (S12D Fig, dashed line). No strong outliers were observed for the rest and movement periods (S12B and S12C Fig) or for any GPi-VLa correlation (Fig 4).

In summary, the simulation results suggest that our empirical data were collected under conditions approximated by the model with parameters in the range of $N \geq 20$ and $c < 0.1$ (i.e., high GPi-to-VLa convergence and low intra-GPi synchrony). Changes of those conditions—for example, increases in synchrony within GPi—may lead to efficient inhibition of thalamus by BG output.

## Discussion

It is often assumed that task-related changes in neuronal activity in BG-recipient regions of thalamus are permitted or caused by the temporal pattern of input from the BG. The physiologic mechanisms most often cited are some kind of gated permission to spike [12; 14; 15] or a triggering of rebound spikes in thalamus through release from sustained inhibition [20; 21; 22]. Recently, Goldberg and Fee [25] demonstrated in the awake songbird that thalamic neuron spiking can be entrained to the ISIs of ongoing pallidal spiking, not only during overt pauses in pallidal firing as proposed by the standard gating model. None of these models have been tested before in the NHP. Here, we sampled single-unit activity simultaneously from connected regions of GPi and VLa thalamus during performance of a reaching task. We searched for evidence consistent with a gating or rebound sculpting of thalamic activity by BG output. Some of our results also bear on the entrainment model. The contrast between our observations against the main predictions of the gating and rebound theories is summarized in Table 3.

We found that peri-movement modulations in discharge were very common in GPi and VLa. Critically, those modulations consisted of increases in discharge more often than decreases both in GPi and in VLa. This finding was supported by two independent analyses: first of the signs (increase versus decrease) of individual response profiles and second of

**Table 3. Overview.**

| | Relationships between GPi and VLa *Hypothesized versus observed* | | |
|---|---|---|---|
| | **Gating** [15; 41] | **Rebound** [18; 22] | **Observed** |
| *Incidence of increase- versus decrease-type responses* | Inverted. GPi increases proportional to VLa <u>decreases</u> | Direct. GPi increases proportional to VLa <u>increases</u> | Direct. Similar rates of increases found in GPi and VLa (see Table 1) |
| *Timing of movement-related responses* | GPi <u>leads</u> VLa by about 2 ms | GPi <u>leads</u> VLa by about 100 ms | GPi <u>lags</u> VLa by about 43 ms (see Fig 3A) |
| *GPi-VLa spike-spike CCFs* | Negative CCF peaks at short latency in subpopulation of pairs | CCF peaks negative at short latency followed by positive at long latency. | CCF peaks are rare (at noise rate). No net bias toward negative or positive peaks (see Fig 4A–4F and Table 2) |
| *GPi-VLa FR correlations (NCs)* | Negative NCs are larger and/or more common | Positive NCs are larger and/or more common | Significant NCs are rare w/ no net bias toward positive or negative relationships (see Fig 4G–4H and Table 2) |
| *VLa FR following burst in GPi firing* | VLa FR decreases at short latency (about 2 ms) | VLa FR decreases following burst onset | Long-lasting (>200 ms) VLa decrease w/ pause part of GPi burst-pause complex (see Fig 5) |
| *VLa FR following pauses in GPi firing* | VLa FR increases at short latency (about 2 ms) | VLa FR increases at short latency | |

Abbreviations: CCF, cross-correlation function; FR, firing rate; GPi, globus pallidus-internus; NC, noise correlation; VLa, ventrolateral anterior nucleus; w/, with

integrated changes in firing across the movement period. It is difficult to reconcile these results with the gating hypothesis without invoking some yet undiscovered mechanism that would make decrease-type responses in GPi, which were in the minority in our observations, more effective at eliciting VLa spikes than increase-type responses are at inhibiting them (e.g., Goldberg and colleagues' [2] "different motor channels" idea).

Both gating and rebound hypotheses predict that task-related changes in GPi activity should begin earlier in time than the neuronal responses they are hypothesized to elicit in thalamus. Contrary to those predictions, we found that onset latencies of GPi responses lagged in time behind those of VLa responses. That was true for a comparison of all responses and, most directly relevant to the hypotheses, for comparisons of GPi decreases versus VLa increases and GPi increases versus VLa decreases. Together, these results bring into question the idea that task-related activity in VLa is generated or permitted by changes in GPi activity. Instead, they buttress previous suggestions [24; 25] that task-related activity in BG-recipient regions of thalamus is generated primarily by some non-BG source (e.g., by glutamatergic inputs from cortex [60; 61]) and that uncorrelated BG inputs have a more subtle influence on thalamic activity than often assumed.

The qualitative form of peri-movement modulations in discharge as well as their timing were comparable between movement directions. Restricting analyses to the preferred direction of each neuron did not change our main findings, stressing the absence of evidence for direction-selective gating or rebound. Compared to rodents [42], the general direction-selectivity of neurons in the BG recorded here was rather weak. That discrepancy may relate to differences in the tasks used for rodents and our NHPs: rodent tasks often [42; 62] (but not always [44; 63]) require large movements of the body in opposing (e.g., left versus right) directions. In contrast, our NHPs performed discrete reaches with one arm toward one of two targets that were positioned only 14 cm apart such that muscle activity for those two reach directions differed only quantitatively (S3 Fig). We cannot rule out the possibility that a task that required selection between categorically different movements would have yielded results more consistent with gating or rebound hypotheses. It is worth noting, however, that our latency results, in which GPi responses began largely after the onset of agonist EMG, agree with many previous studies that used a wide variety of motor tasks performed in NHPs [31; 32; 28; 64] and in rodents [65].

Another approach to test for possible influences of GPi input on VLa activity is to determine the pattern of correlated activity observed in simultaneously recorded GPi-VLa cell pairs. This approach has the potential to elucidate the nature of cell-to-cell communication and how it differs between task conditions [66]. It is revealing that very few GPi-VLa cell pairs (<6%) showed significant spike-to-spike correlations, and in those few, the correlations were small in magnitude (peak variation in firing rate < 30% of the baseline rate) and independent of task period (i.e., during rest or movement periods). Moreover, the whole population of cross-correlations was similar to a control distribution with jittered spike times. We used simulated Poisson spike trains with known underlying correlations to rate the sensitivity of our method and excluded pairs with sparse data that did not allow for detection of small correlations. The result differs markedly from the common assumption, as predicted by gating and rebound hypotheses, that cross-correlations for connected GPi-VLa cell pairs will be strongly negative. It also differs from the strong negative cross-correlations observed by Goldberg and Fee in the songbird BG-thalamic circuit [25]. Our simulations illustrate how the absence of strong cross-correlations in our data may be accounted for by the anatomical convergence, in mammals, of inputs from numerous GPi neurons onto individual VLa neurons [10; 36] as compared with the 1:1 pairing in the songbird of very strong calyceal-type synaptic contacts from single pallidal axons onto an individual thalamic neuron [35].

Proper evaluation of the correlation results discussed above requires consideration of how likely it was for our recordings to encounter synaptically connected GPi-VLa cell pairs. Even though recordings were restricted to regions of the GPi and VLa that were likely to be connected (i.e., regions responsive to stimulation of arm M1), single units were sampled at random from within those regions. The likelihood of recording from connected pairs depends on the detailed anatomy of GPi projections into VLa. Axons of individual GPi neurons terminate in multiple dense glomerule-like clusters in the VLa, up to 10 of which are distributed widely across the VLa [11; 36; 67]. Within each cluster, large multisynapse boutons contact primarily the somata and proximal dendrites of multiple thalamocortical projection neurons [10; 11; 36]. Thus, although exact quantification of the degree of GPi-to-VLa divergence has yet to be performed, it is clear that individual GPi neurons diverge to contact numerous thalamic neurons distributed across the VLa. This anatomic arrangement should markedly improve our chances of encountering connected GPi-VLa pairs by random sampling. The paucity of evidence for connected GPi-VLa cell pairs in our cross-correlation results implies either that the degree of GPi-to-VLa divergence is more sparse than what the anatomy suggests or that the influence of individual GPi cell firing on the recipient VLa neuron was far more subtle in our paradigm than what current theories would predict.

An influence of GPi inputs on VLa activity might also be evident in slow trial-to-trial covariations in the firing rates observed within GPi-VLa cell pairs (noise correlations). If a gating mechanism dominated GPi-VLa communication, then the majority of significant noise correlations would be expected to be negative and/or the overall distribution might be biased toward negative correlations. Noise correlations in our data were occasionally significant (5% of pairs at rest and 7% of pairs during movement), but these were composed of balanced proportions of positive and negative correlations (Fig 4G and 4H) and the overall distribution of noise correlations did not differ from a shuffled control. Significant noise correlations can be produced by a variety of mechanisms other than direct monosynaptic connectivity, which include, most obviously, co-modulation of both neurons in the pair by a third source of input [68].

Bursts and pauses in GPi activity are prolonged neurophysiologic events likely to have more profound effects on postsynaptic neurons than the effects of single spikes [54; 69]. Most important here, a burst of inhibitory GPi input to a thalamic neuron followed by a pause in firing should be an ideal stimulus to trigger rebound-type spiking—if, that is, the rebound mechanism is in effect. As others have described previously [54], we found that spontaneous bursts in GPi firing during periods of attentive rest are often followed by pauses. However, these burst-pause events in GPi neurons were coupled with small yet sustained reductions in mean VLa firing rate, thus opposite of what the rebound mechanism predicts. Critically, those reductions in firing rate occurred at around 119 ms after burst onset and were thus much slower than what would be expected for the synaptic transmission (compare to Fig 1B: VLa was inhibited few milliseconds after GPi stimulation). The lack of VLa rebound responses following GPi pauses in our trials involving successful movements dovetails with the lack of motor impairments found in Cav3.1 knockout mice that lacked thalamic rebound firing [22]. Moreover, both the timing and sign of the observed VLa rate changes were inconsistent with predictions of the gating hypothesis. The observed co-occurrence of VLa firing-rate reductions with GPi burst-pause complexes may reflect large-scale properties of the BG-thalamo-cortical network, similar to those invoked previously to explain the detailed structure of bursts and pauses in pallidal activity [54; 70]. Regardless of that, our results are not consistent with straightforward interpretations of gating or rebound models, both of which hypothesize that thalamic activity is strongly determined by BG output.

Although our results are restricted to a well-learned movement task and rest, which both involve low synchrony of BG output [56; 59] (S12 Fig), effective thalamic inhibition or

excitation may be possible in the presence of BG synchrony. To demonstrate this reasoning, we simulated a simplified version of the recorded circuit (Fig 6). Our simulations confirm that in the presence of anatomical convergence from BG to thalamus and low synchrony of BG output, thalamus is only weakly affected by the BG. In contrast, synchrony of BG output can lead to efficient short-latency inhibition of thalamus, even in the presence of strong convergence. We therefore suggest that BG-output correlations can be a powerful modulator of BG influences on thalamus, which may be exploited under specific behavior conditions (e.g., during reward-based learning) and may be a factor in the pathophysiology of BG disorders [56; 57].

To our knowledge, this is the first study of single-unit activity sampled simultaneously from the GPi and VLa. These results, though novel, are consistent with many previous observations. Past between-studies comparisons observed that task-related increases in firing are more prevalent than decreases both in BG-output neurons [28; 29; 30; 31; 32] and in VLa thalamus [33] (69%); [17] (83%); [34]. The latencies of task-related activity in BG-output neurons [28; 31; 32] also appeared to lag in time behind those in VLa [17; 33; 34]. Our results are consistent with several past reports that movement-related modulations in GPi activity begin after the onset of activity in the agonist muscles but before the onset of overt movement [28; 30; 31].

In addition, task-related changes in VLa activity were unaffected by temporary inactivations of the GPi [24], even though the background firing rate of VLa neurons increased during those inactivations. More recently, Goldberg and Fee [25] confirmed in the songbird BG-thalamic circuit the paradoxical presence of task-related increases in activity both in BG-output neurons and in BG-recipient thalamus and the persistence of task-related activity in the BG-recipient thalamus following ablation of the BG. The present results are also consistent with the more general observation that inactivations or ablations placed in BG-output nuclei have, at most, minor detrimental effects on the performance of familiar motor tasks both in human patients [71; 72; 73; 74] and in neurologically normal nonhuman animals [24; 75; 76; 77; 78].

Given the lack of strong gating or rebound, how can we explain coactivation of GPi and VLa with movement? Rodent layer 5 pyramidal tract neurons of motor cortical areas innervate both BG and BG-recipient thalamus [79; 80]. Thus, excitatory drive from motor cortex with different delays may explain our observation of strong movement-related modulations in firing rate in both GPi and VLa without evidence for strong direct interactions between the two nuclei. Indeed, electrophysiological evidence indicates an efficient excitatory cortical control of motor thalamus [63; 81], possibly explaining the early activation of VLa with movement onset.

Finally, how do we bring the current results into coherence with other studies that demonstrated strong BG-thalamic effects? For example, two recent studies showed that optogenetic stimulation of BG output had profound effects on thalamic activity [22; 82]. Stimulation produced thalamic firing-rate changes along with differences in licking behavior [82] and was followed (at a lag of approximately 70 ms) by a sharp rebound-like increase in thalamic spiking accompanied by muscle contractions [22]. Based on the timing reported there, any similar post-inhibitory rebound in our data would have been apparent in our GPi burst-pause analysis (Fig 5), yet we saw a decrease rather than an increase in thalamic firing. As shown by our model (Fig 6), the impact of convergent BG inputs to a recipient thalamic neurons depends tremendously on the degree of synchronization in spiking between those BG inputs. Obviously, massed stimulation of BG efferent terminals (e.g., using optogenetic methods as in [22; 82]) will induce a synchronized volley of action potentials in a large fraction of the BG-output neurons. Similarly, macroelectrode stimulation of GPi, which inhibited VLa activity in our animals (Fig 1B), induces a volley of action potentials synchronized across a large population of GPi neurons [83]. Also, rodent work has revealed a potent effect of synchronized compared to uncorrelated activity in BG output neurons [84]. Our simulation (Fig 6) shows how such a synchronized population volley (i.e., high level of pairwise correlation c) will have a much

larger postsynaptic influence on thalamic neurons than that of the highly desynchronized spiking that is typical of a nonperturbed population of BG-output neurons [56; 58; 59] (see also S12 Fig). Note that our observation of very low levels of between-neuron synchrony in GPi is consistent with multiple past studies in neurologically normal animals [59; 85], even during performance of well-learned behavioral tasks [56; 86]. The primary BG-output nucleus in the rodent (SNr) shows similarly low levels of pairwise correlations [87], potentially due to desynchronization by intrinsic inhibitory connections [88].

## Potential limitations and caveats

As discussed above, our results describe the dynamics of randomly sampled pairs of neurons in GPi and VLa, and they do not rule out the possibility that strong interactions, including gating or rebound, exist within tightly focused subcircuits connecting those nuclei. For example, it would be nearly impossible to detect by random sampling the very strong entrainment-like interactions observed in the hyperfocused pallido-thalamic circuit of the songbird [25; 35]. The anatomy of the mammalian GPi-VLa projection, however, suggests a far more branched organization containing a great deal of divergence and convergence [10; 11] in which it should be possible to study connected cell pairs by random sampling from GPi and VLa. At minimum, our results put a low upper limit (i.e., less than 15 in 427, see Table 2) on the probability of finding strong spike-to-spike cross-correlation effects, if any exist, in randomly selected GPi-VLa cell pairs. In addition, our results are inconsistent with the classic gating idea in which a coordinated drop in GPi activity is required to release thalamic activity and subsequent selection of action [40; 41].

In the comparison of onset latencies for GPi and VLa responses, a substantial degree of overlap in the distributions (Fig 3 and S8 Fig) leaves open the possibility for gating-like latency relationships in a minor subpopulation of GPi and VLa neurons. However, gating-consistent relationships of response sign and timing were found in only a small fraction (4%) of all possible combinations of GPi and VLa responses. In addition, inconsistent results for GPi and VLa were found when we tested for relations between response latencies and response magnitudes. Early latency responses in VLa were larger in magnitude than later responses whereas GPi responses did not vary in magnitude as a function of latency (S9 Fig). Furthermore, our study of cross-correlations and noise correlations in simultaneously recorded GPi-VLa pairs did not reveal any indication of gating or rebound mechanisms (Fig 4).

It is also important to acknowledge that we studied GPi-VLa communication during performance of a simple well-learned reaching task. It is possible that the influences of BG output on thalamic spiking could be stronger under more demanding or less-stereotyped behavioral contexts. For example, several lines of evidence suggest that BG-thalamic pathways drive behavioral variability or exploration during motor learning [89; 90; 91]. Other studies suggest BG involvement in the on-line modulation of movement vigor [44; 75] or the urgency to move [45]. A growing number of studies have concluded that the influence of BG output on a behavior becomes less important the more well-learned the behavior becomes [27; 77; 92]. It is possible that, under one or more of those less-stereotyped behavioral contexts, task-related activity in the GPi may adopt characteristics more capable of influencing VLa activity (e.g., larger firing rates, shorter latency responses to inputs, or more synchronization of spiking between GPi single units).

## Conclusion

In conclusion, we found no evidence consistent with the idea that BG-output discharge gates thalamic discharge ("classic gating hypothesis," [12]). The most likely alternative is that both

pallidal and thalamic discharge may be driven by a third source [24; 25]. Layer 5 projections from motor cortex are a reasonable candidate for shared excitatory drive [63; 80; 81]. To the extent that they have been compared, all BG-thalamic projections in mammals appear to share similar anatomy and physiology [10]. Because of that, the present results have important implications for BG-thalamic communication in all functional circuits (e.g., in associative, oculomotor and limbic functional circuits [38]), not just the skeletomotor circuit. Our results are compatible with the idea that BG outputs may counterbalance cortical drive to thalamus. For example, increases in BG output may modulate or constrain the magnitude of thalamic changes in discharge, perhaps as a consequence of recent reward history [2]. Finally, subtle changes in BG-output synchrony may be a potent way to scale the effectiveness of inhibitory drive to thalamus.

## Materials and methods

### Data collection

**Ethics statement.**   All aspects of animal care were in accord with the National Institutes of Health Guide for the Care and Use of Laboratory Animals, the PHS Policy on the Humane Care and Use of Laboratory Animals, and the American Physiological Society's Guiding Principles in the Care and Use of Animals. We made all efforts to provide excellent animal care and to alleviate unnecessary discomfort. All surgical procedures were carried out under general anesthesia, using sterile techniques in approved surgery areas. Analgesics were used postoperatively to minimize discomfort or pain. All experimental protocols were performed in strict accordance with the National Institutes of Health Guide for the Care and Use of Laboratory Animals and were reviewed and approved by the University of Pittsburgh IACUC before the studies began (ACUC protocol number 18093682).

**Animals and task.**   Two monkeys (*Macaca mulatta*; G, female 7.1 kg; I, female 7.5 kg) were used in this study at the University of Pittsburgh. The animals performed a choice reaction time reaching task that has been described in detail previously [48; 49]. In brief, the animal faced a vertical response panel that contained two target LEDs, positioned 7 cm to the left and right of midline, and associated infrared proximity sensors. The animal's left hand rested at a "home-position" at waist height and equipped with a proximity sensor. The animal was trained to hold the home-position (1–2 s, uniform random distribution) until the right or left LED was lit as a directional "Go" signal (selected in pseudorandom order). The animal was given 1 s to move its hand from the home-position to the indicated target. Once the correct target was contacted, the animal was required to hold its hand at the target for 0.5–1.0 s (randomized) before food reward was delivered via a sipper tube and computer-controlled peristaltic pump. The animal was then allowed to return its hand to the home-position with no time limit. The right hand was restrained in a padded splint at the animal's right side.

**Surgery.**   General surgical procedures have been described previously [49; 75]. The chamber implantation surgery was performed under sterile conditions with ketamine induction followed by isoflurane anesthesia. Vital signs (i.e., pulse rate, blood pressure, respiration, end-tidal $pCO_2$, and EKG) were monitored continuously to ensure proper anesthesia. A cylindrical titanium recording chamber was affixed to the skull at stereotaxic coordinates to allow access to the right globus pallidus and ventrolateral thalamus via a parasagittal approach. A second chamber was positioned over the right hemisphere in the coronal plane to allow chronic implantation of stimulating electrodes in the arm area of primary motor cortex and the decussation of the SCP. The chambers and head-stabilization devices were fastened to the skull via bone screws and methyl methacrylate polymer. Prophylactic antibiotics and analgesics were administered postsurgically.

A second aseptic surgery was performed in one animal (NHP G) to implant chronic subcutaneous electrodes for electromyographic recording [75]. Pairs of fine Teflon-insulated stainless steel wires (AS632 Cooner Wire) were implanted into six muscles of the proximal arm (biceps, triceps, anterior and posterior deltoid, pectoralis and latissimus dorsi). The wires were tunneled subcutaneously to a connector mounted on the animal's cranial implant.

**Localization of stimulation sites and implantation of indwelling macroelectrodes.** To guide an electrical stimulation–based localization of the region of GPi devoted to arm motor control [93] and of the connected region of VLa [34], we implanted stimulation electrodes in the arm-related region of primary motor cortex and in the SCP at its decussation (Fig 1). The anatomic locations of sites for implantation were estimated initially from structural MRI scans (Siemens 3T Allegra Scanner, voxel size of 0.6 mm) using an interactive 3D software system (Cicerone) to visualize MRI images and predict trajectories for microelectrode penetrations [94]. Subsequent microelectrode mapping methods were used to identify the precise chamber coordinates for the implantation.

Custom-built stimulating electrodes were implanted at these sites using methods described previously [95]. Macroelectrodes consisted of two Teflon-insulated Pt-Ir microwires (50 μm) glued inside a short stainless steel cannula with about 0.5 mm of separation between the distal ends of the microwires. Insulation was stripped from approximately 0.2 mm of the distal ends of the microwire to achieve an impedance of approximately 10 kΩ. The electrode assembly was implanted transdurally via the coronal chamber using a protective guide cannula and stylus mounted in the microdrive. In the months following implantation, the location and integrity of macroelectrodes were monitored by comparing the muscle contractions evoked by stimulation through the electrode against what was observed during microelectrode mapping.

**Localization of target regions for recording in GPi and VLa.** The chamber coordinates for candidate regions in GPi and VLa were estimated initially from structural MRIs as described above. Target region localization was then refined using single-unit microelectrode recording in combination with electrical stimulation through electrodes in SCP and GPi (single biphasic pulses <200 μA, 0.2-ms duration at 2 Hz max.; Model 2100, A-M Systems; Fig 1) and proprioceptive stimulation. The target region for recording in GPi was identified by the presence of typical high-firing-rate single units, many of which responded briskly to proprioceptive stimulation of the forelimb [32; 96] and responses to electrical stimulation in the arm region of primary motor cortex [93] (Fig 1C). During localization of the target region in VLa, a macroelectrode was positioned acutely in the GPi. The target region for recording in VLa was identified by the presence of typical thalamic neuronal discharge that (1) responded to GPi stimulation with a short-latency pause in firing [34], often followed by a rebound increase in firing probability, and (2) did not respond to SCP stimulation, which would be indicative of a neuron in VLp, the cerebellar-recipient portion of motor thalamus located immediately posterior to VLa (Fig 1B). Many VLa neurons also responded at short latency to stimulation in primary motor cortex. All subsequent data collection was directed to these target regions of GPi and VLa.

We also performed microstimulation mapping of VLa and VLp (biphasic pulses <200 μA, 0.2-ms duration at 300 Hz; Model 2100, A-M Systems). Consistent with previous reports, stimulation in putative VLa rarely evoked movement whereas stimulation in putative VLp evoked movement often and at low threshold [97; 98]. However, it was possible to evoke movement from some locations close to VLp but identified as VLa according to the localization criteria described above. Thus, results from microstimulation mapping of the thalamus were not used as primary criteria for identification of the VLa/VLp border.

**Recording protocol.** The extracellular spiking activity of neurons in GPi and VLa was recorded using multiple glass-insulated tungsten microelectrodes (0.5–1.5 MΩ, Alpha Omega)

or 16-contact linear probes (0.5–1.0 MΩ, V-probe, Plexon). Data were amplified (4×, 2 Hz–7.5 kHz), digitized at 24 kHz (16-bit resolution; Tucker Davis Technologies), and saved to disk as continuous data.

All recordings were performed with at least one electrode positioned in each of GPi and VLa. When stable single-unit isolation was available from one or more single units in both GPi and VLa, as judged by online spike sorting, neuronal data and behavioral event codes were collected while the animal performed the behavioral task.

During a subset of data collection sessions, EMG activity was collected via either chronically implanted subcutaneous electrodes (NHP G) or electrodes inserted percutaneously immediately before the session (NHP I). The EMG signals were amplified (4×), band-pass filtered (100–5,000 Hz), digitized (6,104 Hz), rectified, low-pass filtered (500 Hz), and then down-sampled to 1,017 Hz.

### Offline analysis

**Behavior.**   Task performance was screened to exclude error trials and outliers in task performance. Reaction times reflected the time interval between LED lighting and subsequent offset of the home-position proximity detector. Movement durations reflected the time interval between detected departure from the home-position and detected arrival of the hand at the target. Outliers in reaction time or movement duration were defined as values >6× the median absolute difference away from the mean (Matlab TRIM).

**Spike sorting and detection of peri-movement discharge.**   The stored neuronal data were high-pass filtered (Fpass: 300 Hz, Matlab FIRPM) and thresholded, and candidate action potentials were sorted into clusters in principal components space (Off-line Sorter, Plexon). Clusters were accepted as well-isolated single units only if the unit's action potentials were of a consistent shape and could be separated reliably from the waveforms of other neurons as well as from background noise throughout the period of recording. Times of spike occurrence were saved at millisecond accuracy.

Single units were accepted for further analysis if they met the following a priori criteria. A minimum of 10 valid behavioral trials was required for all task-based analyses. (Among the single units studied, the actual minimum number of trials collected—16—was greater than this a priori minimum.) The minimum firing rate, mean across the whole period of recording, was 30 Hz for GPi units and 1 Hz for VLa units.

We tested for peri-movement changes in single-unit spike rate using a standard method [49] that was modified to improve the sensitivity to firing-rate decreases through use of different estimates of unit activity for the detection of increases and decreases in discharge. For increases, we used a standard SDF, which correlates directly with a neuron's mean instantaneous firing rate. For decreases, however, we used a function that reflects a unit's instantaneous ISI [99], which scales with the reciprocal of a neuron's instantaneous spike rate. Use of the ISI function avoided a potential insensitivity for the detection of decreases in SDFs due to floor effects, which would be particularly problematic for low-firing-rate neurons such as those in VLa. (By definition, the minimum value for an SDF is zero spikes/s regardless of the duration of a pause in firing, whereas an ISI function can reliably represent arbitrarily long pauses in firing.) SDFs were constructed by convolving a unit's spike time stamps (1-kHz resolution) with a Gaussian kernel (σ = 25 ms). ISI functions were calculated as a millisecond-by-millisecond representation of the current time interval between successive single-unit spikes smoothed (Matlab CONV) using a 25-ms Gaussian kernel. Across-trial mean SDF and ISI functions aligned on the time of movement onset were constructed separately for valid behavioral trials to left and right targets.

The detection algorithm then tested both SDF and ISI activity functions for significant positive deviations from a control rate within a 700-ms window that started at the median time of target LED onset relative to the time of movement onset (i.e., within a time period that encompassed both reaction time and movement duration for our animals). The threshold for significance was defined relative to the mean and SD of values from a pre-trigger control period (a 700-ms window that ended at the median time of target LED onset) after any linear trend in the mean activity function from that period was subtracted. A movement-related change in firing rate was defined as a significant elevation from the control mean activity that lasted at least 70 ms (e.g., Fig 2A, solid vertical lines; $t$ test; point-by-point comparisons at 1-ms resolution of one sample versus control period mean; omnibus $p < 0.001$ after Bonferroni correction for multiple comparisons). Any such elevations in the SDF were classified as increases in discharge whereas elevations in the ISI function were classified as decreases in discharge. Note that this approach enabled detection of biphasic changes (e.g., an increase followed by a decrease).

To test for potential biases in the response detection algorithm, we generated simulated data with imposed responses of different sizes and then measured the sensitivity of our algorithm to detection of those simulated responses. For each single unit in our empirical database, we generated simulated SDF and ISI activity functions (for detection of increases and decreases, respectively) based on that unit's pre-trigger control period mean ($\mu$) and SD ($\sigma$). Each simulated activity function was 1,500 ms long. For each 25-ms interval of the first 1,000 ms, values were chosen from a normal distribution matching the experimental $\mu$ and $\sigma$. For the period 1,000–1,500 ms, a simulated response was imposed by selecting values from a normal distribution with mean $\alpha\mu$ and SD $\sigma$, where $\alpha$ reflects the change in activity expressed as a fraction of baseline. These values were then interpolated using Matlab's cubic spline interpolator resulting in a simulated activity function that matched both the statistics and the qualitative features of the empirical data. One hundred such SDF and ISI activity functions were created for each single unit and level of $\alpha$ (at intervals of 0.05 between $\alpha = 0.0$ and 3.0). The detection algorithm was applied to all simulated activity functions, and the fraction of responses detected was quantified as a function of $\alpha$, single-unit type (GPi versus VLa), and response sign (increases versus decrease). See S5 Fig and associated caption for a summary of the results from this simulation.

For each significant movement-related change detected, we used two independent approaches to estimate the time of onset (i.e., the latency). The first standard approach [49; 95] simply took the earliest significant time bin yielded by the detection algorithm described above. To ensure that the standard approach did not provide biased results, we also applied a second approach which defined onset as the time at which the mean activity function crossed a threshold corresponding to 10% of the maximum change in rate relative to the control rate (as defined above).

The accuracy of the latency estimation algorithm was tested by applying it to simulated responses that had known onset latencies but otherwise matched the statistics of empirically observed responses. For each increase-type response detected, we created a simulated increase-type response by imposing onto that unit's baseline (±SD) firing rate a trapezoidal increase in firing rate that matched the magnitude and onset slope of the empirical response. Likewise, for each decrease-type response detected, we created matching simulated ISI changes from the baseline ISI (±SD). The latency estimation algorithm described above was then applied to these simulated increase- and decrease-type responses. This analysis revealed a slight nonsignificant bias in the algorithm toward detecting GPi responses earlier than VLa responses (S7 Fig). Note that this bias is the converse of the actual difference in latencies observed between GPi and VLa.

To investigate the possibility that response latency results differed depending on a unit's preferred direction, we tested for significant differences in response magnitude as a function

of movement direction and then identified the neuron's preferred direction. First, we tested for significant differences in a neuron's firing rate for movements to left and right targets. Trial-by-trial spike counts from a 300-ms window starting at the unit's earliest detected response onset were compared between left- and right-target trials. If a unit's spike rate differed significantly between the two movement directions ($p < 0.005$; Matlab RANKSUM), then the unit's "preferred direction" was defined as the direction for which spike rate during the same 300-ms window differed (i.e., either increased or decreased) the most from baseline firing rate.

**Spike and rate correlations.** To estimate the level of fast coordination between spike times in GPi and VLa, we computed CCFs. Spike time series were kept uncut ("whole recordings") or cut trial by trial into 0.5-s-long windows aligned to the time of movement (−0.2 to 0.3 s relative to detected movement onset) or to the pre–go cue rest period (1.5 to 1 s before detected movement onset). GPi units with average firing rates below 30 Hz, VLa units with firing rates below 1 Hz, as well as all units with a total spike count of less than 100 within all relevant trials were excluded. For all other pairs recorded simultaneously in GPi and VLa, the normalized CCF

$$CCF(\tau) = \frac{\sum_{s=1}^{n} y(t_s + \tau)}{n \; \overline{y(t)}}$$

was computed for time lags $\tau \leq 200$ ms, taking advantage of zero-padding, and averaged over all trials of a recording. $t_s$ denotes the s = 1,...$n$ spike times of the GPi neuron, $y(t)$ the binary spike time series of the VLa neuron, and $\overline{y(t)}$ the average VLa firing rate. Surrogate time series were generated by randomly jittering spike times within intervals of 20 ms as suggested by Amarasingham and colleagues [100]. This kind of surrogate data left local firing rates unchanged while removing spike synchrony on a time scale of 20 ms or shorter and thereby serving as a negative control for spike synchrony. Average CCFs of surrogate data were subtracted from both trial averaged CCFs for each GPi-VLa unit pair as well as from all control CCFs. All final CCFs were rated by the absolute value of the maximum deviation from zero in the time interval [0,10] ms ([−5,5] ms for CCFs between GPi neurons) after smoothing with a 2-ms moving average filter.

Additionally, to control for false negative findings, we simulated Poisson spike trains with known correlations. For each recorded pair, 400 simulated spike trains were matched to the firing rates of GPi and VLa neurons and to peak correlations of value $p$ for very long recording times. By subsampling those simulated spike trains at the length of our recordings, we observed simulated distributions of correlations around the underlying correlation $p$. This procedure allowed us to determine the minimum reliably detectable correlation $c_0$ for each pair. $c_0$ was defined as the lowest $p$ for which 95% of the distribution of simulated correlations was larger than 95% of the control distribution. Thus, $c_0$ can be used to rate the sensitivity of correlation detection in each pair. If $c_0 > 0.3$, the pair was excluded from the analysis. Moreover, the approach allowed for estimation of upper bounds $c_u$ of the measured correlations. $c_u$ was defined as the lowest $p$ for which 95% of the simulated correlations were larger than the measured correlation. $c_u$ therefore sets an upper limit to the correlations in our system.

Next, we tested whether correlations in trial-by-trial variations in firing rates ("noise correlations") between GPi and VLa discharge were present. We computed spike counts in 500-ms bins within the same rest and movement periods as used for CCFs. If the total spike count in a time bin across all trials of a unit was lower than 10, the bin of this unit was excluded. Spike counts were z-scored and trials with a score >3 were removed from further analysis, as described by Liu and colleagues [101]. Separately for each pair and each of the two targets,

correlations were then computed across bins of spike counts. Simultaneous modulations of firing rates that occur consistently across movements are thus not reflected in the noise correlations. Instead, only trial-by-trial variations of rate contribute. Randomly shuffling trials within each recording served as surrogate data.

**Analysis of bursts and pauses.** The classic gating hypothesis states that any increased BG output, regardless of its relation to movement timing, can attenuate thalamic spiking [13]. Here, we investigated the influence of GPi bursts and pauses on VLa spiking during rest. Bursts were detected with a "surprise" method developed by Legéndy and Salcman [102] and implemented by Wichmann and Soares [54]. The surprise value was defined as S = −log(P), where P is the probability that the distribution of ISIs within the candidate burst is from a Poisson distribution. Only bursts with a surprise value of 5 or larger, with at least three spikes and an intraburst firing rate of at least twice the baseline firing rate, were considered. Likewise, pauses were defined as ISIs of at least 100 ms with a minimum surprise value of 5. All bursts and pauses that were detected during the time period when the task was performed (from 0.4 s before detected movement onset until 0.8 s after return to the home key) were excluded. As movement periods were associated with strong modulations in firing rate (see Fig 2), reliable burst and pause detection during these periods was not possible.

VLa spike trains were convolved with a Gaussian density of standard deviation $\sigma$ = 10 ms. We then averaged all epochs of VLa spiking from 0.8 s before to 0.8 s after GPi burst onset and called this average spike train a BTA. Each simultaneously recorded GPi-VLa pair thus led to one BTA. We used the same analysis with alignment to burst end to determine BOTAs and with alignment to pause onsets for PTAs. Some pairs included few GPi bursts or low baseline firing rates in VLa, impeding the detection of burst or pause influences. To avoid including such noisy data, we excluded GPi-VLa pairs with noisy pre-burst baselines, defined as the average VLa activity 800–50 ms prior to the respective event (GPi burst onset, offset, or pause): if the difference between the 2.5th and the 97.5th percentile of this baseline was larger than 60% of the absolute average baseline activity, the respective GPi-VLa pair was neglected. Hence, only pairs with a rather constant, predictable baseline were included in the analysis.

All obtained TAs of each type (BTAs, BOTAs, and PTAs) were averaged to compute a population average TA of that type. We also evaluated whether some VLa units showed a significantly high modulation after simultaneously recorded GPi bursts or pauses. Detection thresholds were set to the 2.5th and 97.5th percentile of the baseline before each event. If the average TA within 0–100 ms after the event crossed one of the thresholds, the TA was assigned to be "decreasing" or "increasing," respectively. Finally, we computed a population average of all cross-correlations between GPi pause and GPi burst offset times, both smoothed with a Gaussian density of standard deviation $\sigma$ = 10 ms.

**Statistical testing.** For each analysis relating to cell-pair interactions (Figs 4 and 5, Table 2), we computed surrogate data as described above. Permutations were done 400 times and each set of shuffled data was processed identically to unshuffled data. The resulting 400 surrogate data sets were then used as a control distribution of which the 2.5th and the 97.5th percentile were taken as limits of the 95% CI. For analyses that involve multiple comparisons, the CIs were shifted such that in total, 5% of shuffled controls became significant for any comparison.

Differences in distributions were tested by comparison of Kolmogorov-Smirnov (KS) statistics. We computed the one-sided KS statistic comparing the empirically obtained distribution to 399 control distributions ("test statistic") as well as the one-sided KS statistic of each of the 400 control distributions compared to the remaining 399 control distributions (400 "control statistics"). If the test statistic was larger than the 95th percentile of the control statistics, we concluded that the obtained distribution was significantly right-shifted relative to the control distribution.

**Simulations.** Custom Matlab code was written to generate $N$ ($N = 5, 20, 100$) simulated GPi spike trains of duration 100 s, with ISIs stochastically selected from a gamma distribution, with firing rate 70 Hz for each train, and with a specified level c of spike time correlation between each pair of trains (c = 0, 0.1, 0.2, 0.3, 0.4). Spike times from the collection of all GPi trains were converted into a single synaptic conductance time series. Specifically, each spike was convolved with an exponential kernel with decay time constant 5 ms, scaled to have total area 7 units. The conductance time series computed from all $N$ GPi neurons, multiplied by a factor of 0.03 so that individual inputs near resting potential induce membrane variations of 1–2 mV and then divided by $N$ for normalization, was taken as input in the simulation of 100 s of activity of a model TCN used in past studies [103], performed in the freely available software XPPAUT [104]. TCN spike times were computed as times of voltage crossing through a threshold of −20 mV. Using the XCOV function in Matlab, pairwise cross-correlations were computed between the TCN spike times and those of five randomly selected GPi spike trains for each trial. A total of 20 cross-correlations, from four separate simulations, were averaged for each ($N$,c) pair. From each averaged cross-correlation, the value of maximum magnitude was computed and the time lag at which this maximum occurred was determined.

Average cross-correlations in the interval [0,10 ms] were compared to a linear approximation of the simulations motivated as follows. Let $s_j(t)$ be a representation of the spike train of GPi neuron $j = 1, \ldots, N$. For a discrete time representation, this is a binarized spike train; in continuous time, it is a sum of Dirac delta functions. The synaptic conductance to the single VLa neuron induced by the whole GPi population can be written as

$$g(t) = K * \sum_{j}^{N} J \, s_j(t)$$

where $*$ denotes convolution and $K(t)$ is a kernel representing the postsynaptic conductance waveform. To keep the total conductance fixed as $N$ changes, the synaptic weights were chosen as $J = b/N$ for a constant $b < 0$. The cross-correlation between a single GPi spike train and the total synaptic conductance is the cross-correlation between the two time series where one has been translated in time by lag $\tau$. Omitting the explicit time translation, we can write the GPi-VLa cross-correlation at any fixed lag $\tau$ as

$$\frac{cov(s_k, g)}{\sigma^2}$$

$$= \frac{1}{\sigma^2} cov(s_k, K * \sum_{j=1}^{N} J \, s_j(t))$$

$$= \frac{b \, K}{\sigma^2 N} * \sum_{j=1}^{N} cov(s_k, s_j)$$

$$= \frac{b \, K}{\sigma^2 N} * (\sum_{j \neq k}^{N} cov(s_j, s_k) + cov(s_j, s_j))$$

$$= \frac{b \, K}{N} * ((N-1)c + 1)$$

$$= a \frac{((N-1)c + 1)}{N}$$

where $\sigma^2$ is the autocovariance of each $s_j$, $c = cov(s_j, s_k)/\sigma^2$ is the cross-correlation between GPi spike trains at time lag $\tau$, and $a = b \int_{-\infty}^{\infty} K(t) dt$ is a constant. The cross-correlation between model GPi spike trains and synaptic conductance is mapped nonlinearly to the cross-

correlation between GPi spike trains and the VLa spike train. However, when correlations are small, a linear approximation to the transfer of cross-correlation from input to output is accurate [105; 106]. Hence, the cross-correlation between a model GPi spike train and the VLa spike train at any fixed lag $\tau$ is approximately proportional to $\frac{N-1}{N}c + 1/N$ at least when correlations are weak. The linear approximation was fitted to the data by a least squares fit.

## Supporting information

**S1 Fig. The locations of all GPi and VLa single units included in the database (black tick marks) are plotted on parasagittal sections at 1-mm intervals separately for animals G and I.** Green bars indicate the locations of SCP-responsive VLp neurons. Gray bars indicate the locations of activity characteristic of the reticular nucleus of the thalamus. Line drawings of nuclear boundaries were taken from a standard atlas that was then warped to align with the structural MRIs and microelectrode mapping results from individual animals. GPi, globus pallidus-internus; SCP, superior cerebellar peduncle; VLa, ventrolateral anterior nucleus; VLp, ventrolateral posterior nucleus.
(TIF)

**S2 Fig. Both animals performed the behavioral task in a highly stereotyped fashion with short reaction times and movement durations.** (A) Reaction times did not differ significantly between the two animals (NHP G versus NHP I) or between the two reach directions (left versus right target location). (B) Movement durations were longer for reaches to the right target than to the left target. NHP G moved more slowly overall compared with NHP I. Data and code to reproduce this figure can be found in *https://doi.org/10.5061/dryad.0cfxpnvxm* (FigS2.m). NHP, nonhuman primate.
(TIF)

**S3 Fig. Reach-related modulations in muscle activity began well in advance of the mechanically detected onset of movement.** Rectified low-pass filtered EMG from proximal arm muscles was collected during a subset of data collection sessions ($n$ = 667 and 476 trials in NHPs G and I, respectively). Signals from each muscle were averaged across trials separately for reaches to left and right targets (blue and green traces, respectively). The width of each trace reflects the SEM. The earliest reach-related modulation in EMG in both animals consisted of a reduction in triceps resting activity followed soon thereafter by increases in two agonist muscles (anterior deltoid and biceps; red vertical dashed lines), which occurred at similar premovement timing in both animals (−123 and −135 ms in NHPs G and I). For reference, red tick marks at the bottom of each panel indicate the times on individual trials of go-cue presentation and target touch at the end of the reach. EMG from pectoralis was not available for NHP I because of poor signal quality. Data and code to reproduce this figure can be found in *https://doi.org/10.5061/dryad.0cfxpnvxm* (FigS3.m). EMG, electromyography; NHP, nonhuman primate; SEM, standard error of the mean.
(TIF)

**S4 Fig. Peri-movement activity of individual single units from GPi and VLa (left and right columns, respectively) reflecting each of the four basic forms of response (rows A-D).** The panel for each single unit shows, in overlay, a mean spike-density function (black, left y-axis) and a mean interspike-interval function (gray, right y-axis), both constructed from the same underlying spike train. The spike-density function was used to test for increases in firing rate relative to pre–go cue baseline activity (linear trend ± CI, sloped yellow lines solid and dotted, respectively). The onset time of significant increases in firing are indicated by vertical yellow lines. The interspike-interval function was used to test for peri-movement decreases in firing,

again relative to baseline activity (linear trend ± CI, sloped blue lines). The time of onset of significant decreases in firing are indicated by vertical blue lines. Note the presence of significant increases and decreases in firing for single units with activity classified as polyphasic. Data and code to reproduce this figure can be found in *https://doi.org/10.5061/dryad.0cfxpnvxm* (FigS4.m). GPi, globus pallidus-internus; VLa, ventrolateral anterior nucleus.
(TIF)

**S5 Fig. Performance of the response detection algorithm at detecting simulated increases versus decreases in firing rate.** Curves reflect the mean (±SD) fraction of increases and decreases detected as a function of the simulated response size (expressed as percent of baseline firing rate). For GPi, the fraction of simulated responses detected was very similar for increases and decreases (red and blue curves, respectively) independent of the response size. For VLa, the algorithm was more effective at detecting decreases than increases when the simulated change in rate was relatively small (<100% of baseline). That bias disappeared for larger response sizes (>100% of baseline). Box plots show the distributions of response in our recorded data plotted separately for increases and decreases for each cell type. Data and code to reproduce this figure can be found in *https://doi.org/10.5061/dryad.0cfxpnvxm* (FigS5.m). GPi, globus pallidus-internus; VLa, ventrolateral anterior nucleus.
(TIF)

**S6 Fig. Relationship between neural responses and movement directions.** (A) Population-level peri-movement activity did not differ between directions of movement. Population mean spike-density functions (±SEM) compiled separately for movements to left and right targets. (B) The distribution of all possible pairings of the four response types in individual neurons for left and right directions of movement. Large fractions of neurons in GPi and VLa (color scale) had the same type of response for both directions of movement (denoted by colors in squares located on the diagonals of the matrices). (C) Integrated changes in firing rate during the peri-movement period plotted for individual neurons for movements to left (abscissa) versus right (ordinate) targets. The polarity and magnitude of integrated changes was in general correlated between the two directions for both GPi and VLa units. Nonetheless, the peri-movement activity of many units differed significantly between left and right reaches (red symbols; 53% and 31% of units in GPi and VLa, respectively). (D) The polarities of integrated firing-rate changes for left- and rightward movements were distributed similarly for neurons in GPi and VLa. (E) The overall proportions of peri-movement responses classified into the four response forms. Although the exact proportions differed somewhat between GPi and VLa populations, the overall distribution showed a similar pattern for GPi and VLa populations. Data and code to reproduce this figure can be found in https://doi.org/10.5061/dryad.0cfxpnvxm (FigS6_S8.m). GPi, globus pallidus-internus; SEM, standard error of the mean; VLa, ventrolateral anterior nucleus.
(TIF)

**S7 Fig. Lack of bias in response latency estimation when the detection algorithms were applied to simulated GPi and VLa responses.** By design, the simulated responses began at time zero but otherwise matched the individual metrics of each empirically observed response (baseline rate, baseline variability, response magnitude, and slope of response onset). The figure follows the conventions of Fig 3. Detected onset times in GPi and VLa did not differ significantly. Data and code to reproduce this figure can be found in *https://doi.org/10.5061/dryad.0cfxpnvxm* (FigS7.m). GPi, globus pallidus-internus; VLa, ventrolateral anterior nucleus.
(TIF)

**S8 Fig. Details on cumulative distributions of onset latencies.** (A-C) Cumulative distributions of onset latencies restricted to responses in the preferred direction of units with peri-

movement activity that was significantly directional. (D-F) Cumulative distributions of onset latencies as defined by the alternate, 10% of maximum, method. The figure follows the conventions of Fig 3. (A) Comparisons of latencies of all peri-movement changes detected in GPi neurons (blue) and VLa neurons (purple). VLa responses precede GPi by a median of 28 ms (**$p < 0.001$ rank sum test). (B) Response onset latencies of VLa increases (purple) lead GPi decreases (blue) by a median of 28.5 ms (ns $p > 0.05$ rank sum test). (C) Response onset latencies of VLa decreases (purple) lead GPi increases (blue) by a median of 29.5 ms (*$p < 0.05$ rank sum test). (D) Comparisons of latencies of all peri-movement changes detected in GPi neurons (blue) and VLa neurons (purple). VLa responses precede GPi by a median of 41 ms (**$p < 0.001$ rank sum test). (E) Response onset latencies of VLa increases (purple) lead GPi decreases (blue) by a median of 65 ms (**$p < 0.001$ rank sum test). (F) Response onset latencies of VLa decreases (purple) lead GPi increases (blue) by a median of 13.5 ms (ns $p > 0.05$ rank sum test). Data and code to reproduce this figure can be found in *https://doi.org/10.5061/ dryad.0cfxpnvxm* (FigS6_S8.m, Fig2_3_S8_S9.m). GPi, globus pallidus-internus; ns, not significant; VLa, ventrolateral anterior nucleus.
(TIF)

**S9 Fig. Relation between response magnitudes and latencies is inconsistent between GPi and VLa.** In GPi (blue), response magnitudes did not differ significantly as a function of onset latencies for either increase- or decrease-type responses (top and bottom scatterplots, respectively). In VLa (purple), response magnitudes were larger for early-onset responses than for late responses. Insets: values for rho and significance from Spearman rank correlation tests. Data and code to reproduce this figure can be found in *https://doi.org/10.5061/dryad. 0cfxpnvxm* (Fig2_3_S8_S9.m). GPi, globus pallidus-internus; VLa, ventrolateral anterior nucleus.
(TIF)

**S10 Fig. Significant cross-correlations.** (A-C) Histograms of significant signed peak correlations. (D-F) Example CCFs achieving significance. Vertical scale bars for each example indicates CCF = 0.2. (G-I) Scatterplots of peak correlation values and confidence intervals. Data and code to reproduce this figure can be found in *https://doi.org/10.5061/dryad.0cfxpnvxm* (Fig4_S10to13.m). CCF, cross-correlation function.
(TIF)

**S11 Fig. Sensitivity analysis for CCFs.** (A-C) Cumulative histograms of measured peak correlations (black) before exclusion of pairs with $c_0 > 0.3$, minimum reliably detectable correlations $c_0$ (red), and upper bounds of the measured correlations $c_u$ (green). The cutoff $c_o > 0.3$ is indicated by dashed lines. (D-F) Scatterplots of measured correlations versus $c_o$. Significant correlations are marked red. Data and code to reproduce this figure can be found in *https://doi.org/ 10.5061/dryad.0cfxpnvxm* (Fig4_S10to13.m). CCF, cross-correlation function.
(TIF)

**S12 Fig. CCFs between GPi units recorded from different electrodes.** The figure follows the conventions of Fig 4. Dashed lines in (D) indicate the average CCF after exclusion of outliers. Data and code to reproduce this figure can be found in *https://doi.org/10.5061/dryad. 0cfxpnvxm* (Fig4_S10to13.m). CCF, cross-correlation function; GPi, globus pallidus-internus.
(TIF)

**S13 Fig. Scatterplots of VLa and GPi rates for the three largest noise correlations.** Data and code to reproduce this figure can be found in *https://doi.org/10.5061/dryad.0cfxpnvxm*

(Fig4_S10to13.m). GPi, globus pallidus-internus; VLa, ventrolateral anterior nucleus.
(TIF)

**S14 Fig. Example GPi burst offset–triggered average firing rate in VLa (mean ± SEM).**
Burst offsets correspond to time zero. Data and code to reproduce this figure can be found in
https://doi.org/10.5061/dryad.0cfxpnvxm (Fig5_S14.m). GPi, globus pallidus-internus; SEM,
standard error of the mean; VLa, ventrolateral anterior nucleus.
(TIF)

## Acknowledgments

We thank Lisa Nieman-Vento and Patrick Rice for their contributions to animal care.

## Author Contributions

**Conceptualization:** Jonathan E. Rubin, Robert S. Turner.

**Data curation:** Daisuke Kase, Andrew Zimnik.

**Formal analysis:** Bettina C. Schwab, Daisuke Kase, Marcello G. Codianni, Robert S. Turner.

**Funding acquisition:** Jonathan E. Rubin, Robert S. Turner.

**Investigation:** Daisuke Kase, Andrew Zimnik, Robert Rosenbaum, Jonathan E. Rubin, Robert
S. Turner.

**Methodology:** Bettina C. Schwab, Robert Rosenbaum, Marcello G. Codianni, Jonathan E.
Rubin, Robert S. Turner.

**Project administration:** Robert S. Turner.

**Software:** Bettina C. Schwab, Robert Rosenbaum, Marcello G. Codianni, Jonathan E. Rubin,
Robert S. Turner.

**Supervision:** Jonathan E. Rubin, Robert S. Turner.

**Validation:** Robert S. Turner.

**Visualization:** Bettina C. Schwab, Robert S. Turner.

**Writing – original draft:** Bettina C. Schwab.

**Writing – review & editing:** Andrew Zimnik, Robert Rosenbaum, Jonathan E. Rubin, Robert
S. Turner.

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
