## [Editor Report · Decision Letter 0]

13 May 2020

Dear Dr Schwab, 

Thank you for submitting your manuscript entitled "NEURAL ACTIVITY DURING A SIMPLE REACHING TASK IN MACAQUES IS COUNTER TO THEORIES OF BASAL GANGLIA-THALAMIC COMMUNICATION" for consideration as a Research Article by PLOS Biology.

Your manuscript has now been evaluated by the PLOS Biology editorial staff, as well as by an Academic Editor with relevant expertise, and I am writing to let you know that we would like to send your submission out for external peer review.

Please re-submit your manuscript within two working days, i.e. by May 15 2020 11:59PM.

Kind regards,

Gabriel Gasque, Ph.D.,

Senior Editor

PLOS Biology

---

## [Decision Letter · Decision Letter 1]

30 Jun 2020

Dear Dr Schwab,

Thank you very much for submitting your manuscript "Neural activity during a simple reaching task in macaques is counter to theories of basal ganglia-thalamic communication" for consideration as a Research Article at PLOS Biology. Your manuscript has been evaluated by the PLOS Biology editors, by an Academic Editor with relevant expertise, and by three independent reviewers. You will note that reviewer 2, Joshua Dudman, signed his comments. We were also expecting a fourth reviewer, who is now very overdue. If this last referee belatedly submits the review, I will forward it to you.

In light of the reviews (below), we are pleased to offer you the opportunity to address the comments from the reviewers in a revised version that we anticipate should not take you very long. We will not be able to make a decision about publication until we see your revised manuscript and your response to reviewers. Please also make sure to address the data and other policy-related requests noted at the end of this email.

We expect to receive your revised manuscript within two weeks.

**IMPORTANT - SUBMITTING YOUR REVISION**

In addition to the remaining revisions and before we will be able to formally accept your manuscript and consider it "in press", we also need to ensure that your article conforms to our guidelines. A member of our team will be in touch shortly with a set of requests. As we can't proceed until these requirements are met, your swift response will help prevent delays to publication.

*Copyediting*

*Published Peer Review History*

*Early Version*

*Submitting Your Revision*

Sincerely,

Gabriel Gasque, Ph.D., 

Senior Editor

PLOS Biology

DATA POLICY:

Note that we do not require all raw data. Rather, we ask for all individual quantitative observations that underlie the data summarized in the figures and results of your paper. For an example see here: http://www.plosbiology.org/article/info%3Adoi%2F10.1371%2Fjournal.pbio.1001908#s5

These data can be made available in one of the following forms:

Regardless of the method selected, please ensure that you provide the individual numerical values that underlie the summary data displayed in the following figure panels: Figures 2BC, 3A-C, 4A-H, 5A-D, 6C, S2AB, S4, S5A-E, S6, S7A-F, S8, S9A-I, S10A-F, S11A-H, S12A-C, and S13.

Please also ensure that each figure legend in your manuscript include information on where the underlying data can be found and ensure your supplemental data file/s has a legend.

Reviewer remarks:

Reviewer #1: This study examined the relationship between basal ganglia output and activity in the ventral thalamus in monkeys performing a reaching task. The authors tested currently popular hypotheses including gating and rebounding accounts, and found that these cannot explain the experimental data. Connected pairs of GPi and VLa neurons were analyzed and it was found that neurons in both areas tend to increase firing at the time of movement, and VLa activity often precedes GPi activity.contrary to predictions of the gating and rebounding theories. This is a thoughtful and solid study. The experiments are carefully conducted and the discussion is scholarly and thorough. The results, while not entirely novel, are the first from identified GPi and VLs neurons in monkeys to my knowledge. For the most part, I find the data and arguments convincing, but I have a few suggestions.

1. In the old primate literature, there were claims that basal ganglia activity follows rather than precedes movement onset. This is not the case here. Could the authors comment on this issue? Does the current finding suggest that the basal ganglia may initiate movement?

2. GPi also projects to brainstem/midbrain areas. What is the role of these target regions in the reaching movement? Do the authors expect a different pattern of results? Would gating or rebound be more relevant in the non-thalamic targets? 

3. Movement direction does not appear to modulate the GPi/VLa activity studied here. But previous work has shown strong effects of movement direction in basal ganglia. For example, more recent work (e.g. Barter et al 2015) showed that in rodents there are antagonistic cell groups defined by movement direction, and that basal ganglia specify instantaneous position coordinates. Also, might movmenets other than reaching explain some of the results here, for example small head movements.

4. There is strong agreement between current findings and that by Goldberg and Fee in songbirds. the model proposed by Goldberg and Fee suggested that high frequency entrainment occurs at high levels of glutamatergic inputs when pallidal 'pauses' are no longer necessary for thalamic spiking. What are the relevant candidates for this strong glutamatergic drive? The authors briefly mention 'a third source,' but perhaps more discussion of this issue could be helpful. 

5. The following statement should be qualified, as there is no indication that inactivation or ablation is complete: "inactivation or ablation of BG outputs have, at most, minor detrimental

effects on the performance of familiar motor tasks both in human patients [Svennilson et al. 33 1960; Baron et al. 1996; Cersosimo et al. 2008; Obeso et al. 2009] and in neurologically-normal

non-human animals [Desmurget & Turner 2008; 2010; Piron et al. 2016; Horak & Anderson 1984; Inase et al. 1996]."

Reviewer #2, Joshua Dudman: In the current submission the authors make a very detailed and careful analysis of recordings from the GPi (BG-output) and Vla (BG-recipient motor thalamus) to test what has been for ~30 years a canonical model to explain the role of basal ganglia in the function of forebrain motor circuits. Specifically, the idea that BG output disinhibits motor thalamus activity to select (by gating) a specific action remains a dominant model. However, as the authors note there have begun to be alternative models articulated. Although the data that underlie this dominant model are in fact rather limited and motivated primarily by relatively simple arguments (strong anatomical projection, high baseline activity of an inhibitory projection neuron, etc.). Nonetheless, I am sympathetic to the authors (implicit) belief here that a lot of careful analysis must be marshaled in order to rule out these existing models due to the persuasiveness of the simple arguments noted above. This paper is thus quite important in my mind because it is clearly the best and most careful assessment of the physiological predictions of such a model to date. I also should note that there is good reason such a comparison has not been made previously - it is very challenging to do well and the experiments here in my estimation have indeed been executed very well. My comments are primarily directed towards trying to improve the manuscripts accessibility to a broader audience that I fear could get occasionally lost in some more detailed points inspired by the very careful look being taken here. 

Some general points that may be worth considering:

Although the text discusses other possible sources of common drive of motor thalamus and BG output as mysterious there is clearly some anatomy that can be described and cited to buttress the point and perhaps demystify a bit. Higher order thalamic nuclei like Vla are characterized by L5 and L6 feedback output from motor cortex. These L5 outputs to motor thalamus are pyramidal tract neurons that also project to multiple levels within BG. Thus, there is some clear anatomical basis for the possibility of a dominant, shared excitatory drive from cortex that could modulate both VLa and GPi. 

At times I thought the key point that the absence of clear 'gating' like disinhibition does not mean that there is no functional consequence of this pathway got lost a little. It is returned to in the modeling and discussion at the end. However, if I interpret the data correctly this can be stated more succinctly in a few places. For example, at the end of the intro I thought a clearer statement about the resolution might be useful compared with:

" Our results suggest that, at least during performance of well-learned tasks in neurologically-normal animals, temporal influences of GPi outputs on VLa activity are subtle and that control over the timing and intensity of both pallidal and thalamic discharge may be dominated by other, possibly cortical, inputs."

I think this sentence highlights an unknown question raised by the data (I have more comments on this point below) whereas there is an important upshot to the results that gets a bit lost here. The authors show that the only way in which GPi output could have a very strong influence on activity in Vla is under the assumption of strong convergence of output projections and a highly correlated burst of activity across the GPi output population. Here, the authors provide strong evidence that in a typical reinforced task - reaching to one of two alternate targets - commonly thought to be BG dependent there is not such a strong correlated modulation of GPi output activity (either positive or negative) and that common drive to thalamus and BG dominate the observed activity. To my opinion that is a key positive take home from this paper that is in the discussion but might aid a reviewer by being moved up earlier in the text (here at the end of the intro) as well.

I also wanted to note that there are some similar points that have been made about BG output in the past that are not referenced here. While I understand the emphasis on GPi and NHPs, the SNr has been studied a reasonable amount in rodents (and also projects to the homologous thalamic nuclei). Here too it was pointed out that due to a lack of correlated modulation it can appear that there is little effect of SNr projections, however, strong synchronous activation reveals a relatively potent effect of this inhibitory pathway, Brown 2014 eLife. This may provide additional support for the discussion about Kim 2017 optogenetic stimulation results and extracellular stimulation results discussed here. There has also in the past been data in NHP (Nevet 2007 JNP) showing little pairwise correlation amongst SNr neurons despite strong inhibitory synapses between these neurons further supporting points in this paper. (Although there is also evidence for strong unitary connectivity and effects on spike timing perhaps more akin to bird data in Goldberg in Higgs 2016 JNP as a counterpoint)

I thought in general the readability of the paper would be improved with some, even modest, description of the task that the monkeys are performing. It does appear in bits throughout the results, but for example, in this sentence " The mean firing rate of most single-units showed small but significant ramps during the start position hold period (i.e., before presentation of the task's go cue; p<0.05, linear regression; 97% and 98% of GPi and VLa cells, respectively)" the reader has no appreciation of what "start position hold period" is yet because there has only been a passing citation to describe the task. I felt that the general implications of the results could be easier to understand if the task we described (just a bit) and the predictions of the getting model clarified in that context.

" In summary, the general distribution of different response types and the predominance of firing-rate increases were similar in GPi and VLa (see Supporting Information S5 E), contrary to the prediction from the gating hypothesis that this relationship would be reciprocal."

The point that the gating model cannot explain the data is very clearly and strongly made with the existing data analysis. Yet, I might also point out one place that didn't receive much discussion which is S5C. The gating model would predict if a given GPi neuron is inhibited for say a R reach then it should be positively or null modulated for L reach ('selecting' by disinhibition the R reach). However, those two quandrants (<0 for one direction and >0 for other direction) have relatively little cells and does not capture the plurality of the data.

Some minor points:

Very difficult to see red tick marks in Figure 2A

Spikes/s (z-score) is a strange description for the colorbar

The title for Figure 5D is confusing and could be clarified. In general the discussion is a little complicated to follow. Perhaps some version of a schematic interpretation of these observations could help a reader?

I believe this is the wrong figure reference for this statement (should be S9) and the rest of the following paragraph " Contrary to the expectation that GPi-to-VLa CCF effects would be predominantly inhibitory (i.e., negative), roughly equal numbers of the CCF peaks detected as significant were positive and negative (8 versus 7, respectively; p=0.79, chi-square test; Supporting Information S8 D-F)." Subsequent paragraph should be S10 not S9 for simulation based estimation of correlation sensitivity. I don't have any substantive issues with the figures, I agree with their conclusions.

Reviewer #3: In the present study, the authors aimed at understanding the mechanism by which the basal ganglia-thalamic (GPi-VLa) pathway communicates information. More precisely, they challenged two theories regarding the way GPi and VLa neurons interact during overlearned behavior, the gating and the rebound theories. These theories make several testable predictions, among which a strong temporal control of thalamic neurons by variations of GPi neuron activities. The authors also considered an alternative hypothesis, the "modulation" hypothesis for which a weaker relationship between task-related activities in GPi and VLa is required.

To this aim, the authors recorded and analyzed single-unit activities sampled (simultaneously or not) from connected areas of GPi and VLa in two rhesus monkeys over-trained to execute reaching movements toward two possible visual targets. 

Using multiple and independent analyses, the authors found a majority of movement-related increases of activity in both GPi and VLa, earlier activity onsets in VLa compared to GPi, weak correlated activities in pairs of GPi and VLa cells, and weak effects of bursts or pauses of GPi neurons spontaneous activity on VLa responses. All these observations are not compatible with the predictions of the gating and/or rebound hypotheses. Finally, using simulations, the authors show that the strength of the GPi-VLa communication depends on the degree of anatomical convergence within the GPi-VLa circuit and on the strength of the spike synchrony within GPi.

Based on these experimental and simulation results, the authors propose that the temporal influence of GPi on VLa activity is weak and that responses in the two structure as well as communication between them might be under control of other inputs. 

General comments

Overall, my opinion about the paper is positive. The study addresses a timely and important problem regarding how operations computed in the basal ganglia influence decision and action-related activity in the cortex, via the thalamus. Even if the study does not allow to definitely answer that question, it indicates that two influential views (the gating and rebound theories) are likely false, at least during overlearned behavior. 

The authors very carefully designed their experiment, especially the methods necessary to control for the connection between the GPi and VLa areas in which neurons were recorded. They also show a very rigorous data analysis flow (including systematic control and redundant procedures as well as production of synthetic data to test their analysis algorithms). As a result, the conclusions of the authors appear very convincing and supported by the data. Moreover, limits and potential caveats are mentioned in the discussion section, which is always a positive point in a paper. The present work is thus very solid, important and well communicated via didactic writing. 

I only have two relatively minor critics that I would nevertheless like to see addressed before publication. 

First, the introduction directly starts with the problematic addressed by the authors (i.e. the connection between the basal ganglia and the thalamus). I believe that one or two broader sentences would help the reader to figure out why this question is important and need investigation. For instance, the authors state that "the connection between the basal ganglia (BG) and its major downstream target, the thalamus, has…" This sentence seems to indicate that the downstream pathway from the BG to the brainstem motor center is less important, which is not true. 

My other concern relates to the modeling work. I had the feeling that this modeling exercise was somehow disconnected from the rest of the paper. I had the opportunity to read a previous version of the paper posted on bioRxiv and this older version did not include the simulations. I thus suspect that the authors were encouraged to add these simulations (maybe to address Kim et al., 2017 apparent conflicting results), which is fine, but I would be more convinced if the connection between their experimental data and this approach was better motivated and the interpretation better described. For instance, it is not clear to me whether the simulation result (and more globally the whole paper) supports the Goldberg et al.'s hypothesis about a modulatory role of the BG on the thalamic activity. If results support the modulatory hypothesis, the title of the paper is misleading because it indicates that the three theories are not supported by the data.

Minor points

* Abstract: The sentence mentioning the simulation ("Instead, simulations…") sounds unclear, especially with respect to the rest of the abstract. A better explanation of the simulation and a clearer connection between the two approaches would improve the last part of the abstract in my opinion. 

* Just like the abstract, I found that the part of the introduction mentioning the modeling work lacks clarity. Especially, the sentence "This insight reveals how our results may be brought into congruence with apparently-contradictory existing observations" does not make much sense to me at this stage of the paper. This is regrettable because the rest of the text is really well written and didactic. 

* Results - p 20: The authors refer to the supporting information S9 and not S8 as it is currently mentioned in the whole paragraph.

* Results - p 21: The authors should have mentioned supporting information S10X instead of S9X. Actually, from supporting information S9, all references to supporting information are erroneous and should be corrected with n+1. 

* Results - p 24: It would be interesting to know whether or not the 119ms separating VLa long-lasting decrease of activity from GPi burst onset is compatible with transmission delays between these two areas. 

* Results - p 26: The formulation "c(N-1)/N+1/N is not clear to me. Could the authors provide more details?

* Methods - p 37: How did the authors controlled monkeys' non acting hand? 

* Methods - p 39: In the paragraph entitled "Recording and stimulation protocol", the stimulation protocol is not described.

---

## [Editor Report · Decision Letter 2]

14 Sep 2020

Dear Dr Turner,

On behalf of my colleagues and the Academic Editor, Alexander Gail, I am pleased to inform you that we will be delighted to publish your Research Article in PLOS Biology. 

Early Version

PRESS 

Kind regards,

Vita Usova

Publication Assistant, 

PLOS Biology

on behalf of

Gabriel Gasque,

Senior Editor

PLOS Biology